# Two decades of flask observations of atmospheric $\delta(O_2/N_2)$, $CO_2$, and APO at stations Lutjewad (the Netherlands) and Mace Head (Ireland), and 3 years from Halley station (Antarctica)

Linh N.T. Nguyen[1], Harro A.J. Meijer[1], Charlotte van Leeuwen[1], Bert A.M. Kers[1], H.A. (Bert) Scheeren[1], Anna E. Jones[2], Neil Brough[2,*], Thomas Barningham[2], Penelope A. Pickers[3], Andrew C. Manning[3], and Ingrid T. Luijkx[4]

[1]Centre for Isotope Research, Energy and Sustainability Research Institute Groningen, University of Groningen, Groningen, the Netherlands
[2]British Antarctic Survey, Natural Environment Research Council, Cambridge, United Kingdom
[3]Centre for Ocean and Atmospheric Sciences, School of Environmental Sciences, University of East Anglia, Norwich, United Kingdom
[4]Meteorology and Air Quality, Wageningen University and Research, Wageningen, the Netherlands
[*]Now at National Institute of Water and Atmospheric Research, Wellington, New Zealand

*Correspondence to*: Linh N.T. Nguyen (n.t.l.nguyen@rug.nl)

**Abstract.** We present 20-year flask sample records of atmospheric $CO_2$, $\delta(O_2/N_2)$ and APO (Atmospheric Potential Oxygen) from the stations Lutjewad (the Netherlands) and Mace Head (Ireland), and a 3-year record from Halley Station (Antarctica). We include details of our calibration procedures and the stability of our calibration scale over time, which we estimate to be 3 per meg over the 11 years of calibration, and our compatibility with the international Scripps $O_2$ scale. The measurement records from Lutjewad and Mace Head show similar long-term trends during the period 2002-2018 of $2.31 \pm 0.07$ ppm $yr^{-1}$ for $CO_2$ and $-21.2 \pm 0.8$ per meg $yr^{-1}$ for $\delta(O_2/N_2)$ at Lutjewad, and $2.22 \pm 0.04$ ppm $yr^{-1}$ for $CO_2$ and $-21.3 \pm 0.9$ per meg $yr^{-1}$ for $\delta(O_2/N_2)$ at Mace Head. They also show a similar $\delta(O_2/N_2)$ seasonal cycle with an amplitude of $54 \pm 4$ per meg at Lutjewad and $61 \pm 5$ per meg at Mace Head, while the $CO_2$ seasonal amplitude at Lutjewad ($16.8 \pm 0.5$ ppm) is slightly higher than that at Mace Head ($14.8 \pm 0.3$ ppm). We show that the observed long-term trends and seasonal cycles are in good agreement with the measurements from various other stations, especially the measurements from Weybourne Atmospheric Observatory (United Kingdom). However, there are remarkable differences in the progression of annual trends between the Mace Head and Lutjewad records for $\delta(O_2/N_2)$ and APO, which might in part be caused by sampling differences, but also by environmental effects, such as North Atlantic Ocean oxygen ventilation changes to which Mace Head is more sensitive. The Halley record shows clear trends and seasonality in $\delta(O_2/N_2)$ and APO, where especially APO agrees well with continuous measurements at the same location made by the University of East Anglia, while $CO_2$ and $\delta(O_2/N_2)$ present slight disagreements, most likely caused by small leakages during sampling. From our 2002-2018 records, we find a good agreement with the Global Carbon Budget for the global ocean carbon sink: $2.1 \pm 0.8$ PgC $yr^{-1}$, based on the Lutjewad record. The data presented in this work are available at https://doi.org/10.18160/qq7d-t060 (Nguyen et al., 2021).

## 1 Introduction

The global carbon cycle is a dynamic system that comprises the exchanges of carbon between various reservoirs and is important for studying human-induced climate change and its impacts (Ciais et al., 2013). Accurate determination of anthropogenic $CO_2$ emissions and their partitioning across different reservoirs plays a vital role in understanding the impact of the remaining atmospheric $CO_2$ mole fraction on climate (Friedlingstein et al., 2020). High-precision atmospheric $O_2$ measurements have been proven to be valuable in quantifying $CO_2$ fluxes in the carbon cycle. By combining the decadal trends of atmospheric $CO_2$ and $O_2$, we can quantify the global land and ocean carbon sinks (Bender et al., 1996; Keeling and Shertz, 1992; Manning and Keeling, 2006; Tohjima et al., 2019). This is because $CO_2$

and $O_2$ cycles are closely coupled – in most processes, there is an anti-correlation in the changes of their mole fraction, except for the oceanic uptake of $CO_2$ (Manning and Keeling, 2006). To quantify the various components of the global carbon cycle, the changes in atmospheric mole fraction of the two species can be used in combination with their stoichiometric exchange ratio (ER), which is the ratio of $CO_2$ and $O_2$ exchanged (consumed/produced) in a process. The ER value varies depending on the process, and is close to 1.1 for photosynthesis/respiration (Severinghaus, 1995) and on average 1.38 for the global mix of fossil fuels (Keeling and Manning, 2014).

There are various techniques to measure atmospheric $O_2$ to high precision, such as interferometry (Keeling, 1988); mass spectrometry (Bender et al., 1994); paramagnetic analysis (Manning et al., 1999); gas chromatography (Tohjima, 2000); vacuum-UV absorption (Stephens et al., 2003; Stephens et al., 2021); and fuel cell technology (Stephens et al., 2007). Despite many improvements to these techniques over the years, it is still very challenging to obtain $O_2$ measurements with high accuracy and precision. This is mainly because the atmospheric background mole fraction of $O_2$ is very high – around $209,392 \pm 3$ ppm (Tohjima et al., 2005) – while the observed variations are at the level of a few ppm. These challenges are magnified further for long-term measurements because of possible small biases, drifts or other changes in the analysers or in the calibration scales. Thus, the sampling procedures and analysing (laboratory) conditions must be monitored and corrected for by a carefully designed use of calibration and reference gas cylinders over the years (Aoki et al., 2021). As a result, there are only a handful of programmes around the globe which are proficient in coupled $CO_2$ and $O_2$ measurements, for example, the network of atmospheric stations maintained by the Scripps Institution of Oceanography (Manning and Keeling, 2006); National Institute of Advanced Industrial Science and Technology (Aoki et al., 2021); National Institute for Environmental Studies (Tohjima et al., 2008); Tohoku University (Goto et al., 2017); University of East Anglia (UEA) (Pickers et al., 2017); and the University of Groningen (van Der Laan-Luijkx et al., 2010). Our laboratory – the Centre for Isotope Research (CIO) of the University of Groningen (RUG) in the Netherlands – has been carrying out flask measurements of $CO_2$ and $O_2$ since the early 2000s from various locations (van Der Laan-Luijkx et al., 2010). Flask sampling for $CO_2$ and $O_2$ has been conducted at Lutjewad (the Netherlands), Mace Head (Ireland), Jungfraujoch (Switzerland) and Halley (Antarctica).

In this paper, we present the $O_2$ and $CO_2$ measurements from flasks collected at Lutjewad (the Netherlands), Mace Head (Ireland), both for the period 2000-2020, and Halley (Antarctica) for 2014-2017. From these measurements, a tracer called Atmospheric Potential Oxygen (APO) (the details of which are given in Sect. 2.5) is calculated. We first describe the measurement sites and the sampling procedure as well as the measurement methods, including the calibration procedure. Then we present the data and discuss the trends and seasonality as well as the quality of the datasets. This paper builds on work previously presented in van Der Laan-Luijkx et al. (2010), Sirignano et al. (2010), and van Leeuwen (2015).

## 2 Methods

### 2.1 Site description

The stations from which our flasks were collected are: Lutjewad Atmospheric Monitoring Station on the northern coast of the Netherlands (53°24'N, 6°20'E) managed by the CIO (RUG); Mace Head Atmospheric Research Station on the western coast of Ireland (53°20'N, 9°54'W) operated by the National University of Ireland's School of Physics and Ryan Institute Centre for Climate & Air Pollution studies; and Halley VI Research Station, at the time of the sampling situated on the Brunt Ice Shelf (75°34′S, 25°30′W), operated by the British Antarctic Survey. Halley station has been relocated later due to that part of the ice shelf breaking off. Figure 1 shows the locations of the three stations.

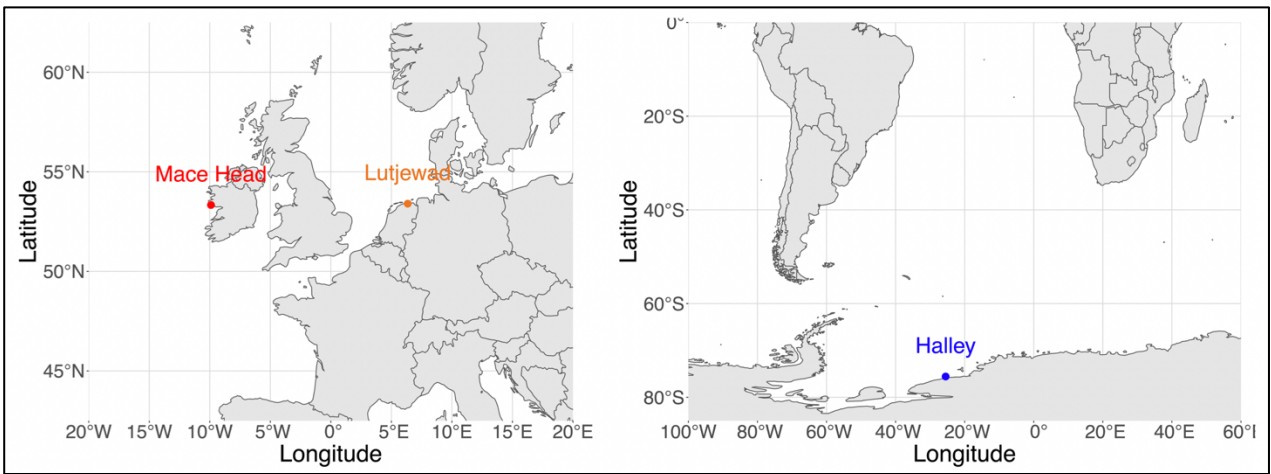

**Figure 1: Left panel: Locations of the Mace Head (red) and Lutjewad (orange) stations. Right panel: Location of the Halley station (blue)**

The Lutjewad station is a "class 2" station in the European Union's Integrated Carbon Observation System (ICOS) network. It comprises a 60-m tall tower, an additional platform of 10-m height, and a laboratory building containing analysers, flask sampling systems, measurement systems and other equipment. The dominant wind direction in the

90 Netherlands is southwest, meaning that the measurements acquired at the Lutjewad station often represent continental air masses influenced by anthropogenic and biogenic sources and sinks (van Der Laan et al., 2010). Otherwise, when the wind comes from the north, the station samples background air that comes from the North Sea and North Atlantic (van Der Laan-Luijkx et al., 2010).

The Mace Head station consists of field laboratories and a 20-m tower for sampling. The dominant wind arriving at the station is westerly from the North Atlantic Ocean, carrying air masses that would not have been considerably affected by regional anthropogenic activities. Air masses from other directions carry contamination from local and continental sources (Derwent et al., 2002; Jennings et al., 1993).

The Halley station is a "Global" station within the World Meteorological Organisation's Global Atmosphere Watch (WMO/GAW) programme, that observes background atmospheric conditions at various locations around the globe. The main Halley station consists of 8 modules that are atop ski-fitted hydraulic legs, within which are the research facilities and living quarters. Air sampling for this project was carried out at the Clean Air Sector Laboratory, which is located 1.5km from the main station in a location that receives minimal contamination from station activities (Jones et

al., 2008). The predominant winds are from the east, bringing background air masses from the South Atlantic sector of the Southern Ocean (60%) or from the continental plateaux (30%). Westerly winds that have passed over the Weddell Sea gyre occur 10% of the time (Barningham, 2018; British Antarctic Survey, 2021).

**2.2 Flask sampling procedure**

At Lutjewad, we employ an automated flask sampling system, hereafter called the autosampler (Neubert et al., 2004).

Air is pumped from the top of the 60-m tower via inlets connected to a series of tubing towards the laboratory building. The inlet is equipped with a Nafion drying tube (MD 110-72-S, Perma Pure, Toms River, New Jersey) so that the incoming air is first partly dried. The flow in the outer side of the Nafion tube is the outlet of the same air sampling system, after the air is dried with the second stage cryogenic dryer in the laboratory to a dewpoint below -45 °C (Neubert et al., 2004). This ensures that, except for water, all constituents have a negligible gradient over the Nafion membrane.

From the inlet, the sampled air is stored in glass flasks via a flask sampling system for further analyses in the CIO laboratories (Neubert et al., 2004). For storing air samples, we use 2.5-litre glass flasks with dip tubes, capped with two high-vacuum valves (Louwers, Hapert, NL) sealed with Viton o-rings (these flasks are also used at Mace Head and Halley). Our autosampler is designed to connect to and fill up to 20 flasks without requiring user intervention, and we can remotely control the opening/closing of the flask valves (via custom-made electric motor actuators) and the filling of samples (via a series including a small diaphragm pump (KNF N811), flow controllers, and magnetic solenoid valves). The autosampler schedule is controlled via custom-made software (written in Delphi programming language), and carries out the sampling procedure automatically, but it can also be operated remotely using software such as VNC or TeamViewer when needed. A normal filling procedure starts with the air stream being cryogenically dried (to a dewpoint of -45°C) and flushed through a flask for at least an hour at 2.5 L min$^{-1}$ before filling the flask slowly so that the sample remains at current atmospheric pressure (to prevent the sample from fractionation and differential permeation through the o-rings caused by a pressure gradient (Sturm et al., 2004)) and moving to the next flask. Individual flasks can be preserved at any time. Samples at Lutjewad are collected under various conditions and time frequencies, but in this paper we present only the data from flasks collected under local background conditions, defined by van Der Laan-Luijkx et al. (2010) as flasks taken while the $^{222}$Radon activity monitored at the station was less than 3 Bq m$^{-3}$ and with a CO mole fraction of less than 200 ppb. This filtering procedure is applied to the dataset after the flasks are analysed.

We employ the same type of flasks, flow rates, and filling pressure (to current atmospheric pressure) at all stations. Due to different setup of the stations, the drying methods are different, and only Halley station has an aspirated inlet.

At Mace Head, flasks are collected once or twice per week via a manually operated system as described by Conway et al. (1994), at 35 m above sea level and mostly during restricted baseline conditions (Bousquet et al., 1996). A sampling sequence starts with the air being pumped from the inlet via a small diaphragm pump (KNF N86KT), into a drying tube packed with magnesium perchlorate, then flushed through the flasks for about 30 minutes at 2.5 L min$^{-1}$ at atmospheric pressure before each flask is manually closed. Also, for Mace Head, only flasks with a CO mole fraction of less than 200 ppb are retained.

At Halley, flasks are collected once a week depending on the meteorological conditions, via a portable manual sampler. This consists of a diaphragm pump (KNF N86), flowmeter, drying agent (magnesium perchlorate), 7µm filter and 3 sampling flasks connected in concession. The air is sampled about 6 metres above the snow surface on the east side of the building via Synflex tubing connected to an aspirated inlet (the details of the aspirated inlet are as described by Blaine et al. (2006)). The system is flushed for about 45 minutes at a flow rate of 2.5 L min$^{-1}$ at atmospheric pressure before each flask is manually closed. The collected samples are stored in insulated aluminium boxes at room temperature until their annual return to the UK on the Antarctic supply ship.

After sample collection, flasks from the three stations are transferred back to our laboratory in Groningen for analysis. Typically, the mole fractions of $CH_4$, CO, $CO_2$ and $O_2$ (reported as δ($O_2/N_2$), see next section) are measured (van Der Laan et al., 2009), and additional analyses such as stable isotopes (for example $^{13}$C and $^{18}$O in $CO_2$) and radiocarbon ($^{14}$C in $CO_2$) are also conducted when required (van Der Laan et al., 2010).

**2.3 CO₂ measurement**

All flask samples are analysed on an Agilent HP6890N gas chromatograph equipped with a Flame Ionization Detector (referred to as HPGC) to determine the mole fractions of $CO_2$, CO and $CH_4$. The HPGC system has a set-up similar to the GC-systems described by Worthy et al. (2003) and van Der Laan et al. (2009). All working standard mixtures (made from dried ambient air) that were used to calibrate the HPGC have been calibrated on the HPGC system at CIO against a suite of 5 primary standards linked to the World Meteorological Organization (WMO) X2007 scale with $CO_2$ ranging between 354 and 426 μmol mol$^{-1}$ (ppm). These primary standards were provided by the Earth System Research Laboratory (ESRL) of the National Oceanic and Atmospheric Administration (NOAA), USA. Since the summer of 2013, working standard gas cylinders were also calibrated for $CO_2$, CO and $CH_4$ mole fractions on a Cavity Ring-Down Spectrometer (CRDS) model G2401-m from Picarro Inc. using the same suite of primary standards. We refer to Chen et al. (2010) for more details on the CDRS technique. The measurement precision and accuracy for flask measurements of $CO_2$ on the HPGC are typically <0.06 ppm and <0.07 ppm, respectively (van Leeuwen, 2015).

All $CO_2$ measurements presented in this paper were originally calibrated against standards on the WMO X2007 scale, and are updated to the WMO X2019 scale (the new scale is explained in details by Hall et al. (2020)).

**2.4 O₂ measurements**

Atmospheric $O_2$ is typically reported as the $\delta(O_2/N_2)$ value. The $\delta(O_2/N_2)$ value of a sample is calculated as the difference between the $O_2/N_2$ ratio of the sample and that of a reference gas (Keeling and Shertz, 1992):

$$\delta(O_2/N_2) = \frac{(O_2/N_2)_{sample} - (O_2/N_2)_{reference}}{(O_2/N_2)_{reference}} \tag{1}$$

Since for natural variations, $\delta(O_2/N_2)$ values are very small, they are usually expressed in "per meg", which is 1/1000 of a per mil, as typically used in the stable isotope community. Atmospheric $O_2$ is reported as $O_2/N_2$ ratio because it is not a trace gas, and its mole fraction is thus affected by changes in other atmospheric constituents such as $CO_2$. Atmospheric $N_2$ is very stable (Keeling et al., 1998), therefore changes in the $O_2/N_2$ ratio would reflect mostly the changes in atmospheric $O_2$ (only in a detailed budget analysis minor $N_2$ variabilities are still considered, as described in Keeling and Manning (2014)). For $\delta(O_2/N_2)$ measurements, we use a Micromass Optima Dual Inlet Isotope Ratio Mass Spectrometer (DI-IRMS). The DI-IRMS analytical technique (which was first developed by Bender et al. (1994)) follows the principles as explained by Keeling et al. (2004). Each measurement comprises sixteen successive switches between sample and reference gases from the respective bellows. After every switch, the pressures of the two bellows are equalized, using a differential pressure meter (GA63, Effa France), subsequently there is an idle period of 120 s before the actual signal is measured for 30 s, to account for the disturbances in the signals caused by the switching of the valves that affect the measurement precision (Sirignano et al., 2010). Due to the sensitivity of the analyser, it is located inside a climate-controlled room in our CIO laboratory. However, it is inevitable that the measurements still drift over time. To correct for the instrumental drifts, we perform frequent calibrations using a suite of reference gas cylinders. These cylinders are calibrated against the international Scripps scale using three primary standard cylinders purchased from the Scripps Institution of Oceanography (SIO), with $\delta(O_2/N_2)$ values ranging from -792 to -254 per meg. Details of the extensive calibration procedure are thoroughly described by van Der Laan-Luijkx (2010) and van Der Laan-Luijkx et al. (2010), and are summarised in Sect. 3.

## 2.5 Atmospheric Potential Oxygen (APO)

Combining highly precise measurements of atmospheric $CO_2$ and $O_2$ can isolate the effects of the oceanic processes, by removing the effects of the land biosphere (Stephens et al., 1998). This is achieved by deriving the tracer Atmospheric Potential Oxygen (APO). The APO value of an air sample is determined by combining its $\delta(O_2/N_2)$ and $CO_2$ measurements (Battle et al., 2006; Gruber et al., 2001; Stephens et al., 1998):

$$\delta APO = \delta(O_2/N_2) + \frac{1.1 \times (CO_2 - 350)}{S_{O_2}}$$ (2)

The value of 1.1 represents the mean $O_2$:$CO_2$ ER of terrestrial ecosystems (Severinghaus, 1995); for the $S_{O_2}$, we take 0.2094, which is the standard atmospheric $O_2$ mole fraction (Tohjima et al., 2005); and 350 is the consensus (arbitrary) reference value to be subtracted from the measured $CO_2$ mole fraction, as defined in the SIO per meg scale conversion for APO (Manning and Keeling, 2006). Therefore, APO is not affected by land biosphere processes and mainly captures the seasonal and long-term air-sea exchange of $CO_2$ and $O_2$, with an influence from fossil fuels combustion, caused by their higher average ER of $\approx 1.4$ (Pickers et al., 2017; Sirignano et al., 2010).

## 3 Calibration of the DI-IRMS

In this section we present the calibration procedure and the stability achieved at our laboratory from 2006 to 2020. The calibration of the measurements made in the 2000-2011 period and reported by van Der Laan-Luijkx et al. (2010) and van Der Laan-Luijkx et al. (2013) are kept intact, and the newly calibrated measurements from 2011 onwards are built on the principles of that work.

### 3.1 The calibration procedure

The DI-IRMS compares the measurement of a sample gas with that of a reference gas (hereby called "machine reference" or "MREF") in a sequence of several switches back and forth ("change overs"). The result of this process is the $\delta(O_2/N_2)$ value of the sample, as presented in equation 1. Each individual measurement is based on seven successive pairs of sample and reference measurements, which are used to calculate seven delta values (equation 1). The seven delta values then go through a filtering process. First, the mean and standard deviation of the seven delta values are calculated. Then, the delta value that is furthest from the mean is marked as a potential outlier. Next, a new mean and a new standard deviation are calculated for the remaining six delta values. If the excluded delta value is more than 2.7 times (equivalent to p = 0.01) the new standard deviation away from the new mean, it is defined as an outlier and removed. This process is repeated to identify and remove a potential second outlier (at most two outliers are removed by this process, otherwise the reliability of the measurement is sacrificed). After removing possible outliers, the remaining delta values are averaged to produce one $\delta(O_2/N_2)$ value per measurement. A flask is typically measured two to three times consecutively, for which we do not find any systematic biases. The final measurement for each flask (as presented in this paper) is the average of the filtered $\delta(O_2/N_2)$ values of these repeated measurements (van Der Laan-Luijkx et al., 2010). The precision of the DI-IRMS for flask measurements varies between 7 and 12 per meg, based on the averaged standard deviation of all flask measurements at Lutjewad and Mace Head flasks, respectively.

To improve the stability of our measurements, we also measure local reference gas cylinders (hereafter called "working tank" or "WT") on the sample side of the DI-IRMS. These WTs are also used to connect between periods of different MREF cylinders, where there may be shifts in the scales of the measurements and thus a scale conversion is required to keep all raw measurements on a comparable scale. The summary of different WTs and MREF cylinders used from 1998 to 2020 is shown in Fig. 2 and Table 1.

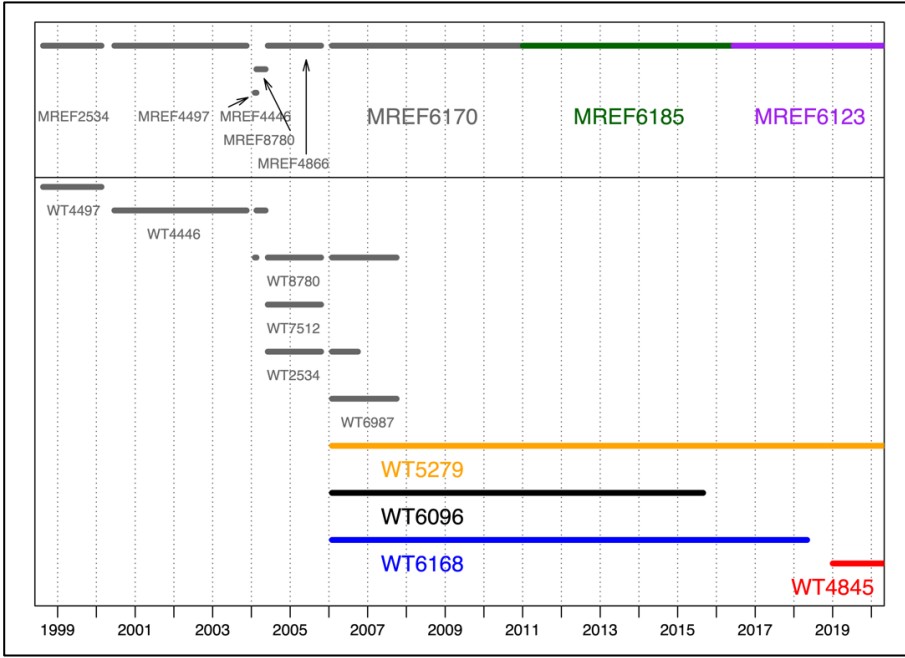

**Figure 2: Summary of the different WTs and MREF cylinders in the 1998 - 2020 period. MREFs are shown along the top, with WTs below. In the case of WTs, there is typically overlap between more than one WT. Periods in grey colour are adapted from the work of van Der Laan-Luijkx (2010).**

To connect the different MREF periods, we first convert all raw measurements (which are the ratios of the raw values to their respective MREF) to our internal 2534 CIO scale. Subsequently, they are converted to the SIO scale. Cylinder number 2534 has been chosen as the baseline for our internal reference scale, because it was the first MREF gas in 1998 and later on was measured as a WT against several other MREF cylinders (Fig. 2). When converting the measurements to the internal CIO scale, we need to take into account the "zero-enrichment" factor: measurements of a WT (on the sample side) against an MREF cylinder (on the reference side) do not produce the same value as when they are measured the other way around (van Der Laan-Luijkx et al., 2010).

In addition to the conversion to our internal CIO scale, the measurements are also affected by instrumental drifts over time. To correct for these drifts, we first divide our long measurement record into several periods, which are defined based on the timing of when the MREF cylinders are changed, and/or apparent fluctuations in the raw data related to, for example, repairs or modifications of the system. In this work, the calibration procedure is carried out for measurements from 2011 onwards, which were divided into seven periods (periods 9-15, Table 1).

**Table 1: Summary of the calibration periods defined in this paper and the corresponding MREF cylinder and WT cylinder numbers most recently used for the calibration of the DI-IRMS. The greyed-out rows are the past cylinders used prior to this work, but included here to demonstrate a complete record.**

| | | Period | | MREF | WTs |
|---|---|---|---|---|---|
| Previous | 1 | 17-08-1998 – 18-02-2000 | | 2534 | 4497 |
| | 2 | 19-06-2000 – 17-11-2003 | | 4497 | 4446 |
| | 3 | 03-02-2004 – 18-02-2004 | | 4446 | 8780 |
| | 4 | 18-02-2004 – 14-05-2004 | | 8780 | 4446 |
| | 5 | 04-06-2004 – 19-10-2005 | | 4866 | 2534 \| 7512 \| 8780 |
| | 6 | 30-01-2006 – 30-12-2006 | | 6170 | 2534 \| 6987 |

| | 7 | 30-01-2007 – 30-12-2007 | 6170 | 5279| 6096 |
| | | | | 6168 | 6987 |
| | 8 | 30-01-2008 – 15-12-2010 | 6170 | 5279 | 6096 | 6168 |
| | 9 | 03-01-2011 – 11-03-2014 | | 5279 | 6096 | 6168 |
| | 10 | 11-03-2014 – 29-08-2015 | 6185 | 5279 | 6096 | 6168 |
| | 11 | 30-08-2015 – 10-06-2016 | | 5279 | 6168 |
| Current | 12 | 11-06-2016 – 05-05-2018 | | 5279 | 6168 |
| | 13 | 06-05-2018 – 01-01-2019 | 6123 | 5279 |
| | 14 | 02-01-2019 – 11-03-2020 | | 5279 | 4845 |
| | 15 | 12-03-2020 to present | | 5279 | 4845 |

These 7 periods were divided into 144 sub-periods (selected based on breaks in the records) which were then individually processed to derive the final corrections for all measurements in those sub-periods. The complete step in transforming the raw measurements of a sample (S) against a current MREF (M) into comparable data is to combine the drift correction with the shift to the CIO scale (R), by using an equation described by van Der Laan-Luijkx (2010):

$$\delta_{S/R} = \left( \left( \delta_{M/R} \right)_{sub-period} + drift \times \frac{days}{365} + 1 \right) \times \left( \delta_{S/M} + 1 \right) - 1 \qquad (3)$$

Where:

- $\delta_{S/R}$ is the $\delta(O_2/N_2)$ value of the sample against the CIO 2534 scale;
- $(\delta_{M/R})_{sub-period}$ is the average $\delta(O_2/N_2)$ value of the MREF cylinder against the CIO scale in a sub-period calculated based on the measurements of all WTs in that sub-period;
- drift is the average drift per day in a sub-period (if any), calculated based on the WT values and days is the number of days at the time of the sample since the start of the sub-period.
- $\delta_{S/M}$ is the $\delta(O_2/N_2)$ value of sample against the MREF cylinder (raw value).

The final step is to transform the $\delta_{S/R}$ value of a sample onto the SIO scale via a linear conversion (shown in Sect. 3.2) using the values of the Scripps primary cylinders measured against the CIO scale. For an extensive and detailed explanation on how to calculate each component of equation 3, we refer to van Der Laan-Luijkx (2010). Figure 3 shows the results for the WTs of the new calibration procedure connected to the previously reported data by van Der Laan-Luijkx et al. (2010).

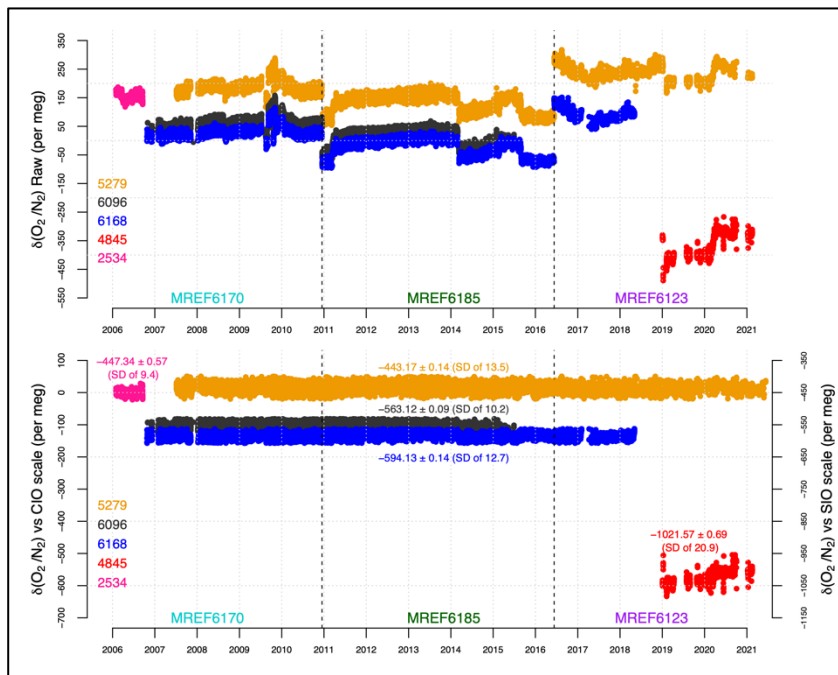

**Figure 3: Measurements of the 3 long-term WTs (5279, 6096, and 6168) for periods 7-15 (Table 1), across the final 3 MREF periods plus a recently added WT (4845). Top panel: raw measurements of the WTs against different MREF cylinders. Bottom panel: measurements of the WTs calibrated and converted to the CIO scale (left y-axis) and against the SIO scale (right y-axis). The values on the plot are the corresponding long-term means and 1-sigma standard errors of the WTs against the SIO scale, and in parentheses are the respective standard deviations. All numbers are in per meg. Visible gaps in the data are due to instrument issues, maintenance or instrument relocation.**

After these adjustments, the measurements of the three long-term WTs (5279, 6096, and 6168) show that all three were simultaneously stable over time. To verify this, we calculated the trends of all three WTs based on their annual averages, and the weighted mean slope amounts to -0.4 ± 0.7 per meg yr$^{-1}$, so not significantly different from zero (Fig. 4). In addition to this outcome, we calculated the year-to-year variability of the WTs, visible in figure 4 as the scatter of the points around the trend lines. We calculated the standard deviations of this scatter (the residuals) around these trend lines, and all three standard deviations were between 2.4 and 4.0 per meg. Therefore, we state our year-to-year stability to be 3 per meg over the 11 years of measurements.

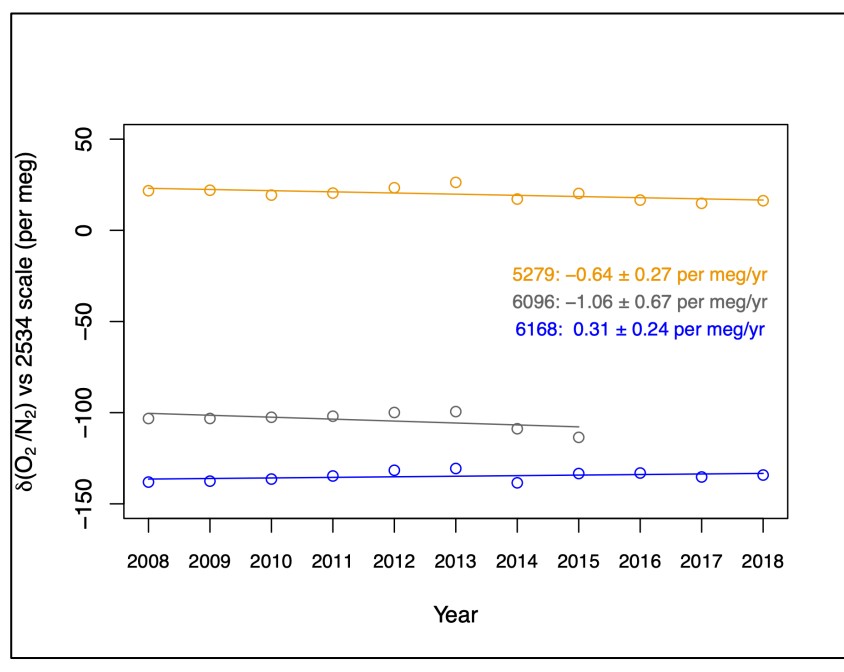

**Figure 4: Annual averages of WTs 5279, 6096, and 6168 (points) from 2008 to 2018, against the CIO scale. The fitted trends (lines) and the values of their slopes are also plotted.**

WT 4845 was recently measured for a relatively short period only, and appeared to be less stable and noisier compared to WT 5279 measured in the same period. It is not clear why this is the case, but it could be due to the fact that the value of this cylinder is very low, suggesting a potential contamination when the cylinder was filled, or a leak in the pressure reducer when it was measured. Thus, WT4845 was not used for the calculations in the calibration procedure, and its measurements are only shown here for completeness.

In addition to their long-term stability, the 3 WTs also showed no systematic drifts across different MREF periods (Table 2). For WT 5279 and WT 6096, there were no significant changes (at least to ±0.3 per meg) between the MREF periods 6170 and 6185, although there was a small decrease of 4.0 per meg in the mean measurement of WT 5279 in MREF 6123 period. For WT 6168, the mean value increased by 3.6 per meg from MREF 6170 to MREF 6185 period, then dropped slightly (by 0.5 per meg) in MREF 6123 period. The stability demonstrated in both long-term measurements and per MREF periods consolidates the quality of our calibration procedure.

Table 2: Comparison of the WTs over 3 different MREF cylinder periods. The values (in per meg) are averaged over the corresponding period, accompanied by the standard errors. The N/A values in the MREF6123 period for WT6096 are due to its discontinuation in this period. The Difference column is calculated by subtracting the values of the old MREF periods from the new ones

|  |  | CIO scale | SIO scale | Difference (CIO scale) |
|---|---|---|---|---|
| WT 5279 | MREF 6170 | 20.9 ± 0.2 | -438.2 ± 0.2 | |
|  | MREF 6185 | 20.7 ± 0.3 | -438.4 ± 0.2 | -0.2 |
|  | MREF 6123 | 16.7 ± 0.2 | -442.3 ± 0.2 | -4.0 |
| WT 6096 | MREF 6170 | -103.4 ± 0.1 | -556.9 ± 0.1 | |
|  | MREF 6185 | -103.1 ± 0.1 | -556.7 ± 0.1 | +0.3 |
|  | MREF 6123 | N/A | N/A | |
| WT 6168 | MREF 6170 | -137.3 ± 0.2 | -589.3 ± 0.2 | |
|  | MREF 6185 | -133.7 ± 0.2 | -585.9 ± 0.2 | +3.6 |
|  | MREF 6123 | -134.2 ± 0.4 | -586.4 ± 0.3 | -0.5 |

## 3.2 Quality check of the Scripps primary cylinders

The final check on the quality of our scale is the regular measurement of the 3 Scripps primary standard cylinders that we purchased from SIO, numbered 7002, 7003, and 7008. These measurements were conducted at least once a year or when there was an additional need for recalibrating e.g. after instrument failure or upgrade. Each measurement period took a different amount of time – some measurements were spread over a couple of days while others were repeated over (or after) a few weeks. From 2007 to 2018, 16 measurement periods were conducted (Fig. 5). The large gap between 2011 and 2014 was due to a lack of funding, and thus of personnel, leading to the situation that the laboratory was understaffed and we could not keep up the measurements of the primary tanks.

In Figure 5, each data point is the mean value over each measurement period and the error bars are the standard deviations. The coloured lines are the overall linear fit of the measured values of the corresponding cylinders (and their associated 2-sigma uncertainties) and the black horizontal lines are the assigned values of the cylinders (determined by

the SIO, updated in 2020). The assigned and measured values of the primary standard cylinders over the whole period are compared in Table 3. The measured values are the weighted means of each cylinder, since each data point is calculated based on different numbers of separate measurements. It can be seen from Figure 5 and Table 3 that cylinder 7008 exhibits a small upward drift over time of $1.4 \pm 0.4$ per meg yr$^{-1}$, whereas the other two remain constant. The ensemble thus suggests that there is no clear systematic error in our scale conversion and calibration procedure. Overall, the SIO primary standards produce a weighted uncertainty of 8.6 per meg in 10 years. To improve the quality of our conversion into the SIO scale, and especially to check the behaviour of cylinder 7008, we are planning to purchase new primary standard cylinders in the future.

The conversion of the CIO scale to the Scripps scale is calibrated using these Scripps primary standards measurements, and in such a way that the ensemble difference between the assigned values and weighted averages of our measurements of three Scripps cylinders is minimised (Mook, 2000):

$$\delta(O_2/N_2)_{SIO} = \delta(O_2/N_2)_{CIO} * 0.999544 + (0.999544 - 1) * 10^6 + 1.4$$

Where $\delta(O_2/N_2)_{SIO}$ and $\delta(O_2/N_2)_{CIO}$ are the $\delta(O_2/N_2)$ values of the SIO and CIO scales, respectively; 0.999544 is the slope with an uncertainty of 0.000008, and 1.4 per meg is the weighted mean offset of the three Scripps primary standards with an uncertainty of 5 per meg (which is thus zero within its uncertainty, as it should).

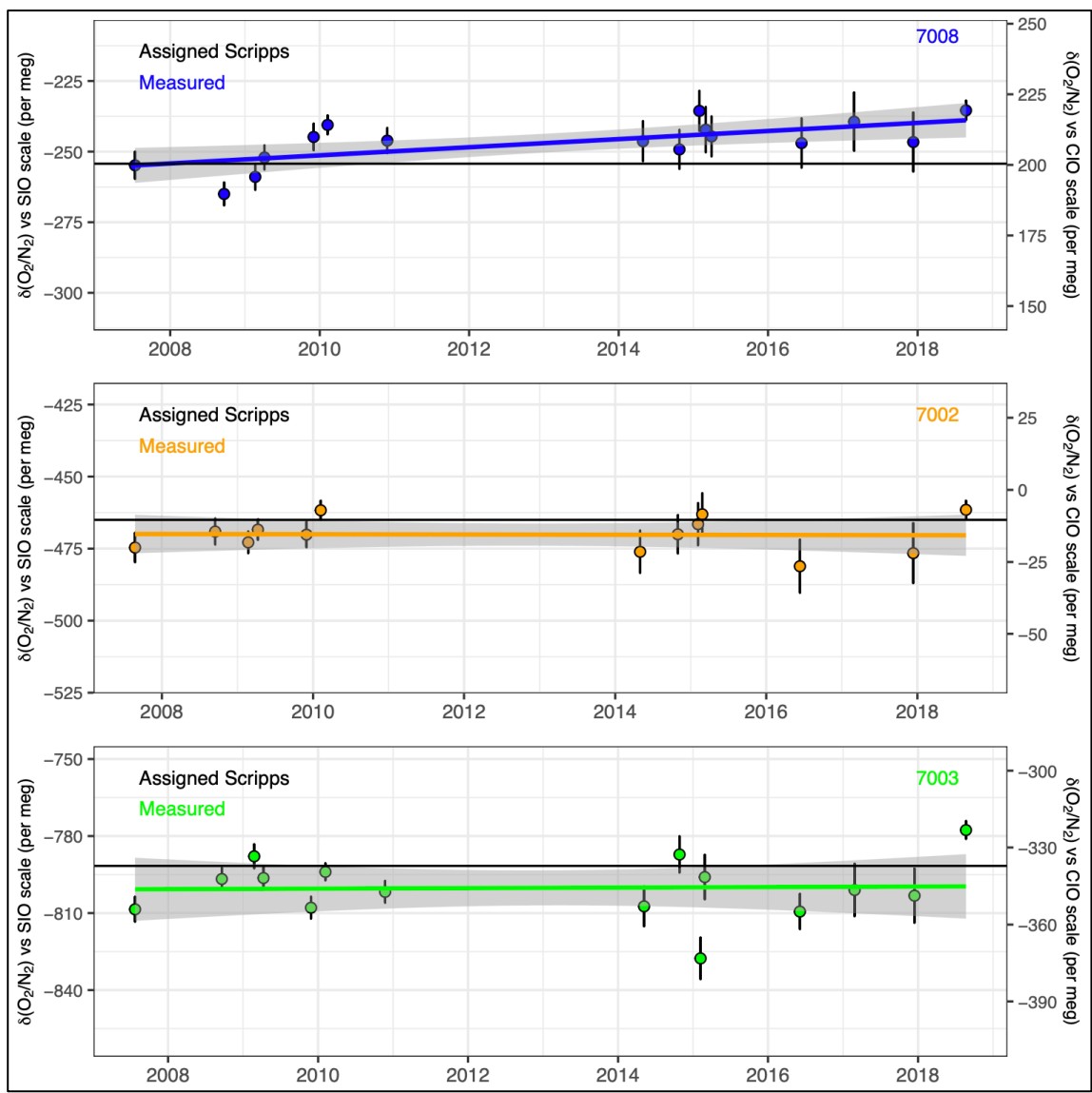

**Figure 5: Scripps primary standard cylinder measurements over time. Each point is the averaged value over a measurement period. Error bars represent 1-sigma standard deviations. Solid horizontal lines are the assigned values**

(black) and the linear least squares fit to the data (coloured) of each cylinder. The grey shading indicates the 95% confidence interval uncertainties of the values.

**Table 3: Comparison of the averaged measured values of the Scripps primary standards against their assigned values in per meg.**

| Cylinder ID | 7008 | 7002 | 7003 |
|---|---|---|---|
| Assigned by SIO | -254.3 | -465.0 | -791.6 |
| Weighted mean measured | -247.3 | -470.3 | -799.0 |
| Standard deviation | 8.0 | 6.0 | 11.8 |
| Standard Error | 1.9 | 2.0 | 2.8 |
| Deviation from assigned | 6.9 | -5.3 | -7.4 |

### 3.3 Inter-comparison programmes

In addition to measuring the primary standard cylinders, the CIO also took part in two inter-comparison programmes involving oxygen measurements: "Cucumber" Intercomparison which was initialised in the European Union's CarboEurope project and coordinated by the UEA (http://cucumbers.uea.ac.uk/); and the Global Oxygen Laboratories Link Ultra-precise Measurements (GOLLUM) programme, also coordinated by UEA (Manning et al., 2015) . These inter-comparison programmes provide an additional tool for checking the internal stability of our measurements, while also linking the oxygen measurements between global laboratories.

The Cucumber programme involves inter-comparison of nine atmospheric species (of which $\delta(O_2/N_2)$ is one) between atmospheric research stations in Europe and a number of laboratories in Europe, USA, Canada, Japan, and Australia. Within the programme, there are seven sets of three cylinders sent around in different rotations. The CIO participated in three rotations, with two involving oxygen measurements (called "Inter-1" and "Euro-3") (University of East Anglia, 2021).

The GOLLUM programme is specifically designed for the inter-comparison of oxygen measurements and involves 10 laboratories worldwide that carry out high-precision atmospheric oxygen measurements. Two sets (named "Bilbo" and "Frodo") of three cylinders are rotated in opposite directions amongst participating laboratories (Manning et al., 2015).

Figure 6 shows the measurements of the Cucumber cylinders (top two panels), the cylinders in the Bilbo and Frodo rotations of GOLLUM (third and fourth panels, respectively) and the measurements of three internal cylinders at CIO: the working tanks 5279, 6096 and 6168 along with the SIO primary standard cylinder 7008 (bottom panel). The measurements of the cylinders in the Inter-1 and Euro-3 rotations are plotted as the difference between the measured values of the cylinders against their own assigned values as originally measured at the Max Planck Institute for Biogeochemistry in Germany in January 2008. These results show that the cylinders in the Inter-1 and Euro-3 rotations were quite variable over time (varying within a range of less than 30 per meg) but in different directions and size, suggesting that there is not a systematic scaling error but rather individual variations between cylinders and/or measurement periods. Due to the individual variations, the overall drifts for Cucumber cylinders is $11 \pm 18$ per meg yr⁻

[1], significantly higher than the WMO network compatibility goal of 2 per meg (World Meteorological Organization,
2018). The lower quality of the measurements (not only in our laboratory) might well be connected to the fact that these
cylinders are not part of a dedicated oxygen comparison programme, so the treatment of the cylinders (for example,
vertical storage and unsuitable pressure reducers) are not of high enough standard for oxygen.

For GOLLUM cylinders, all measurements are also plotted as the difference between the measured values of the
cylinders and their assigned values on the SIO scale. The assigned values for Bilbo, Frodo and SIO cylinders are
determined at the SIO, while those for the WTs are their averaged long-term value measured at CIO on the SIO scale.
Compared to the Cucumber cylinders, GOLLUM cylinders show much less variations between years (varying within a
range of less than 20 per meg), and also significantly smaller overall drift over the duration of the measurements ($4 \pm$
$6$ per meg $yr^{-1}$). However, all 6 cylinders appear to drift in similar direction, suggesting a significant drift in our scale
rather than drifts in these cylinders. The SIO cylinder 7008 also shows similar stability and a general drift in the same
direction as GOLLUM cylinders, whereas the two other SIO cylinder do not (Fig. 5).

Since the cylinders show an inconclusive "drift": INTER-1 and EURO-3 do not show an apparent drift direction; Bilbo
and Frodo present a minor drift similarly to that observed by our SIO cylinder 7008 (while the other 2 SIO cylinders
did not exhibit this behaviour as shown in Sect. 3.2); and our internal WTs all show no overall drifts, we consider our
calibration procedure as sufficient. Recalibration of the SIO cylinders might shed further light on these small
discrepancies, mostly to see if cylinder 7008 has indeed drifted or not.

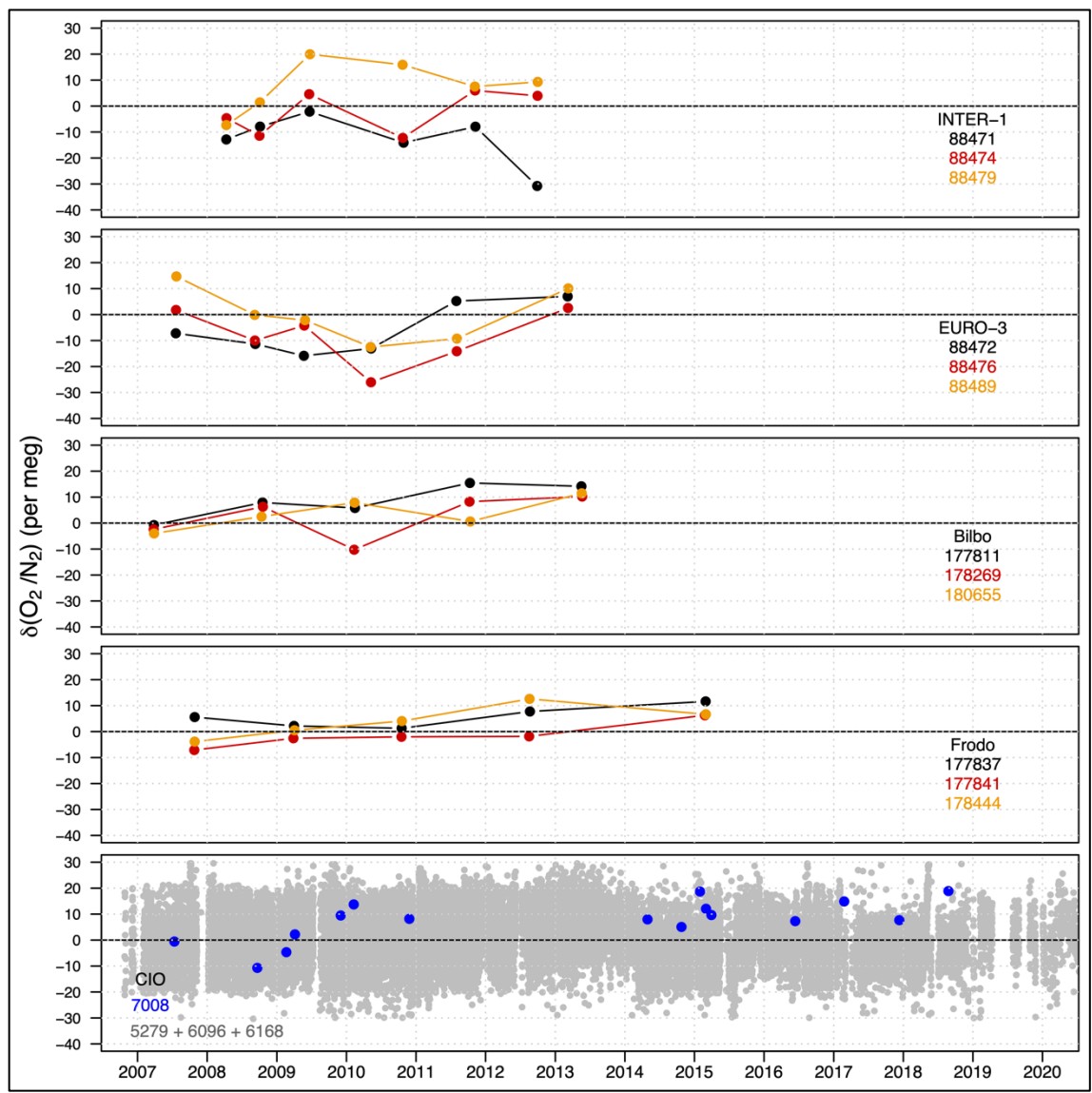

**Figure 6: Cylinders from the Cucumber programme (top 2 panels) along with two sets of three cylinders in the GOLLUM programme (middle 2 panels) and 3 internal CIO cylinders (WT 5279, WT 6096 and WT 6168) and a primary standard cylinder at CIO (SIO 7008) (bottom panel). Each colour represents a different cylinder, and the legends show the corresponding cylinder IDs. The points are the measurements of the cylinders over time, plotted as the difference from their assigned values. For the Cucumber, GOLLUM and SIO cylinders, the assigned values are determined at the SIO, and for the WTs, the assigned value is its long-term average measured at CIO on the SIO scale. Y-axis ranges are identical on all panels.**

### 3.4 Treatment of analysed flask samples

After the calibration and conversion to the SIO scale, the individual flask sample measurements are scrutinized for outliers and background conditions. For this purpose, we perform several iterations of fitting a combination of quadratic and 3-harmonic regression (following similar curve fitting methods applied to time series in NOAA without the use of a digital filtering method (Thoning et al., 1989)) and filtering the outliers from the combined fit. This outlier filtering process uses the robust median absolute deviation (MAD) method (Rousseeuw and Verboven, 2002), in which the MAD value for a dataset is determined by first finding the median of the set, then subtracting the median from each individual value, and finally finding the median of the absolute differences. Measurements that are 3 times the MAD value away from the median of the measurement set are considered outliers and removed. The full principle of the procedure is described by van Der Laan-Luijkx (2010) (though with a different filtering process that was described in Sect. 3.1). In total, after both filtering processes, around 30% of the flasks were excluded from further analyses from Lutjewad samples, 16% from Mace Head samples, and only 6% from Halley samples. The larger fraction of discarded

measurements in the Lutjewad record is related to the sampling process, where we do not specifically only sample air at background conditions, which is the case at Mace Head. For Halley, since it is by design a background station, there are hardly any local sources and sinks, and the wind coming from the continental plateaux only accounts for 30% of the total. The 6% outlier fraction for Halley is a good indication of the fraction of actually failed sampling and/or analysis. The APO values of all stations are calculated from $\delta(O_2/N_2)$ and $CO_2$ measurements (equation 2), when there is information on both species for each flask sample.

In the period prior to 2006, our internal calibration scale was not as well-established as in the later period, due to frequent changes in MREF and WT cylinders, especially in 2004 when there is little information to connect the following period to the first period (as presented in Fig. 2 and Table 1). Next to this, we also only obtained the SIO primary standards in late 2007, so all earlier measurements cannot be directly linked to the SIO scale and have to be converted via the internal CIO scale. The results of this quality check prompt us to exclude the first 2 years from the fits of Lutjewad and Mace Head data so that they are less affected by the problematic period. The last 2 years are also excluded, partly because flask sampling was relatively sparse in those years and this could also introduce biases in the fits, and also because in the period of late 2019 until the whole of 2020, our DI-IRMS experienced detrimental problems that affected the quality of the measurements. After several tests, we decided to establish our fits for Lutjewad and Mace Head based on the years 2002 to 2018.

In summary, in our 20 years of measurements, we have observed an uncertainty on flask measurements of 7 to 12 per meg (based on the averaged standard deviations of the individual flasks collected from Lutjewad and Mace Head), and we have maintained the stability of our internal scale (3 per meg in 11 years) as well as the Scripps primary standards (8.6 per meg in 10 years). Although some drift is observed in one of our Scripps cylinders, the other two have remained stable within uncertainty. The same inconclusive picture emerges from our various sets of cylinders in the inter-comparison programmes. Therefore, we conclude that our calibration process is accurate within the uncertainties mentioned above.

## 4 Flask measurement results

### 4.1 The $CO_2$, $\delta(O_2/N_2)$ and APO records

In this section, we present the long-term flask measurement records (from 2000 to 2020) of Lutjewad and Mace Head, along with a 3-year record from Halley. In general, Lutjewad and Mace Head show similar patterns for $\delta(O_2/N_2)$ and $CO_2$, with some differences in APO variations. Figures 6 to 8 show the $CO_2$, $\delta(O_2/N_2)$, and APO measurements for Lutjewad, Mace Head, and Halley, respectively. The black points illustrate the final, filtered flask measurement values; the coloured lines are the total fit (combined quadratic trend and 3-harmonic seasonal cycles) and the black lines are the trend parts of the total fit. The fit lines are shown for the whole period, but for the fitting process we left the first and last two years out, to make sure that the fit period comprises complete calendar years (from January to December). Otherwise, the beginning and end of the curves can influence the trend part of the fit due to the irregular sampling frequency and other problems, as explained in Sect. 3.4. From the records, the total uncertainties associated with the trends are also calculated, based on a quadratic sum of the uncertainties of the flask measurements and other factors. For $CO_2$, the only other contributing factor is the uncertainty in the trend fit. For $\delta(O_2/N_2)$, and APO, the uncertainties associated with the measurements of the SIO primary standards, our internal scale, the long-term scale conversion between CIO and SIO scales, and the trend fits all contributed to the final uncertainty.

CO2 measurements at Lutjewad and Mace Head show a positive, and increasing trend over 17 years. Due to the quadratic trend fit, the growth of the fitted increase is linear. The trend (given here in ppm $yr^{-1}$ with their 95% confidence interval (CI) uncertainties) in Lutjewad grows from $1.81 \pm 0.10$ ppm $yr^{-1}$ in 2002 to $2.27 \pm 0.03$ ppm $yr^{-1}$ in 2010 and $2.74 \pm 0.10$ ppm $yr^{-1}$ in 2018. These values agree relatively well with the globally averaged values as measured by the NOAA's Global Monitoring Laboratory: $1.86 \pm 0.20$ ppm $yr^{-1}$ in 2002, $1.97 \pm 0.14$ ppm $yr^{-1}$ in 2010, and $2.57 \pm 0.19$ ppm $yr^{-1}$ in 2018 (https://gml.noaa.gov/ccgg/trends/global.html). The values from NOAA are calculated based on a 5-year average around the time marks 2002, 2010 and 2018. In all three periods, the values at Mace Head are also in agreement with those of Lutjewad ($1.86 \pm 0.06$ in 2002, $2.24 \pm 0.02$ in 2010, and $2.63 \pm 0.06$ ppm $yr^{-1}$ in 2018 for Mace Head). When averaging the trends over the 17-year period, both stations show good agreement with each other and with the global average: $2.31 \pm 0.07$ ppm $yr^{-1}$ for Lutjewad, $2.22 \pm 0.04$ ppm $yr^{-1}$ for Mace Head, and $2.1 \pm 0.3$ ppm $yr^{-1}$ for global. The total uncertainty of the trend is 0.07 ppm $yr^{-1}$ for Lutjewad and 0.04 ppm $yr^{-1}$ for Mace Head. The largest contributing factor to the total CO2 long-term trend uncertainty is from the trend fits.

$\delta(O_2/N_2)$ measurements at Lutjewad also show a clear trend that becomes increasingly more negative throughout the 20 years. The trends (reported here in per meg $yr^{-1}$ with their 95% CI uncertainties) in 2002, 2010, and 2018 are $-18.01 \pm 1.17$ per meg $yr^{-1}$, $-20.99 \pm 0.29$ per meg $yr^{-1}$, and $-23.98 \pm 1.17$ per meg $yr^{-1}$, respectively. At Mace Head, we find an unexpected trend: while the trend in CO2 increases, that of $\delta(O_2/N_2)$ becomes less negative ($-22.4 \pm 1.3$ per meg $yr^{-1}$, $-21.2 \pm 0.3$ per meg $yr^{-1}$, and $-20.0 \pm 1.3$ per meg $yr^{-1}$ in 2002, 2010, and 2018, respectively), which is contrary to the expectations of an increasingly negative trend, based on increased fossil fuel consumption over the years, and also different from the measurements at Lutjewad. The lower number of flask samples from Mace Head between 2017 and 2019 makes it difficult to accurately interpret the cause of this change in the trend, and it also affects the determination of a proper fit through the period, potentially leading to inaccuracies in the long-term trend. When averaged over the entire period, however, both stations show almost identical trends: $-21.2 \pm 0.8$ per meg $yr^{-1}$ for Lutjewad and $-21.3 \pm 0.9$ per meg $yr^{-1}$ for Mace Head. The total uncertainty of the trend is 1.3 per meg $yr^{-1}$ for Lutjewad and 1.5 per meg $yr^{-1}$ for Mace Head. The largest contributing factor to the total $\delta(O_2/N_2)$ long-term trend uncertainty for Lutjewad and Mace Head is the uncertainty in the trend fits, with a small effect from the scale stability (of 3 per meg in 11 years). However, at Mace Head the uncertainties in the flask measurements contributed more significantly than those at Lutjewad (12.5 compared to 7.4 per meg, respectively).

The APO trend and seasonality can be determined either from fitting the APO values of the individual flasks themselves, or by combining the trend/seasonal parameters of the $\delta(O_2/N_2)$ and CO2 fits. Both methods yield almost identical results. We present here the results from the first approach. Since APO is calculated from the combination of $\delta(O_2/N_2)$ and CO2 measurements, it shows a combination of the patterns as illustrated in the two species. The APO trend (reported here also in per meg $yr^{-1}$) at Lutjewad does not differ significantly over time, varying from $-9.4 \pm 0.8$ per meg $yr^{-1}$ in 2002 to $-9.31 \pm 0.20$ per meg $yr^{-1}$ in 2010, and $-9.3 \pm 0.8$ per meg $yr^{-1}$ in 2018. In Mace Head, however, the same pattern as $\delta(O_2/N_2)$ is shown for APO: the trend gets significantly less negative throughout the period ($-13.15 \pm 1.20$ per meg $yr^{-1}$ in 2002, $-9.5 \pm 0.3$ per meg $yr^{-1}$ in 2010, and $-5.83 \pm 1.20$ per meg $yr^{-1}$ in 2018). The total uncertainty of the trend is 1.0 per meg $yr^{-1}$ for Lutjewad and 1.3 per meg $yr^{-1}$ for Mace Head, and the largest contributing factors are the same as for $\delta(O_2/N_2)$.

Measurements at Halley station show a similar trend as Lutjewad and Mace Head, where CO2 increases over time while $\delta(O_2/N_2)$ decreases, with much less variability in $\delta(O_2/N_2)$ and CO2 measurements, due to the absence of a terrestrial biosphere influence. The averaged CO2 trend at Halley from 2014 to 2017 is $2.60 \pm 0.20$ ppm $yr^{-1}$, similar to the trends

at Lutjewad and Mace Head in the same period ($2.62 \pm 0.08$ ppm yr$^{-1}$ and $2.53 \pm 0.05$ ppm yr$^{-1}$, respectively). On the other hand, $\delta(O_2/N_2)$ and APO trends at Halley are significantly smaller in size than those at Lutjewad and Mace Head. The $\delta(O_2/N_2)$ trend at Halley over the 2014-2017 period is $-15 \pm 3$ per meg yr$^{-1}$ while at Lutjewad and Mace Head, the trends are $-23.2 \pm 0.9$ per meg yr$^{-1}$and $-20.3 \pm 1.0$ per meg yr$^{-1}$, respectively. For APO, the corresponding values are $-1.4 \pm 2.4$ per meg yr$^{-1}$, $-9.3 \pm 0.6$ per meg yr$^{-1}$, and $-6.7 \pm 0.9$ per meg yr$^{-1}$.

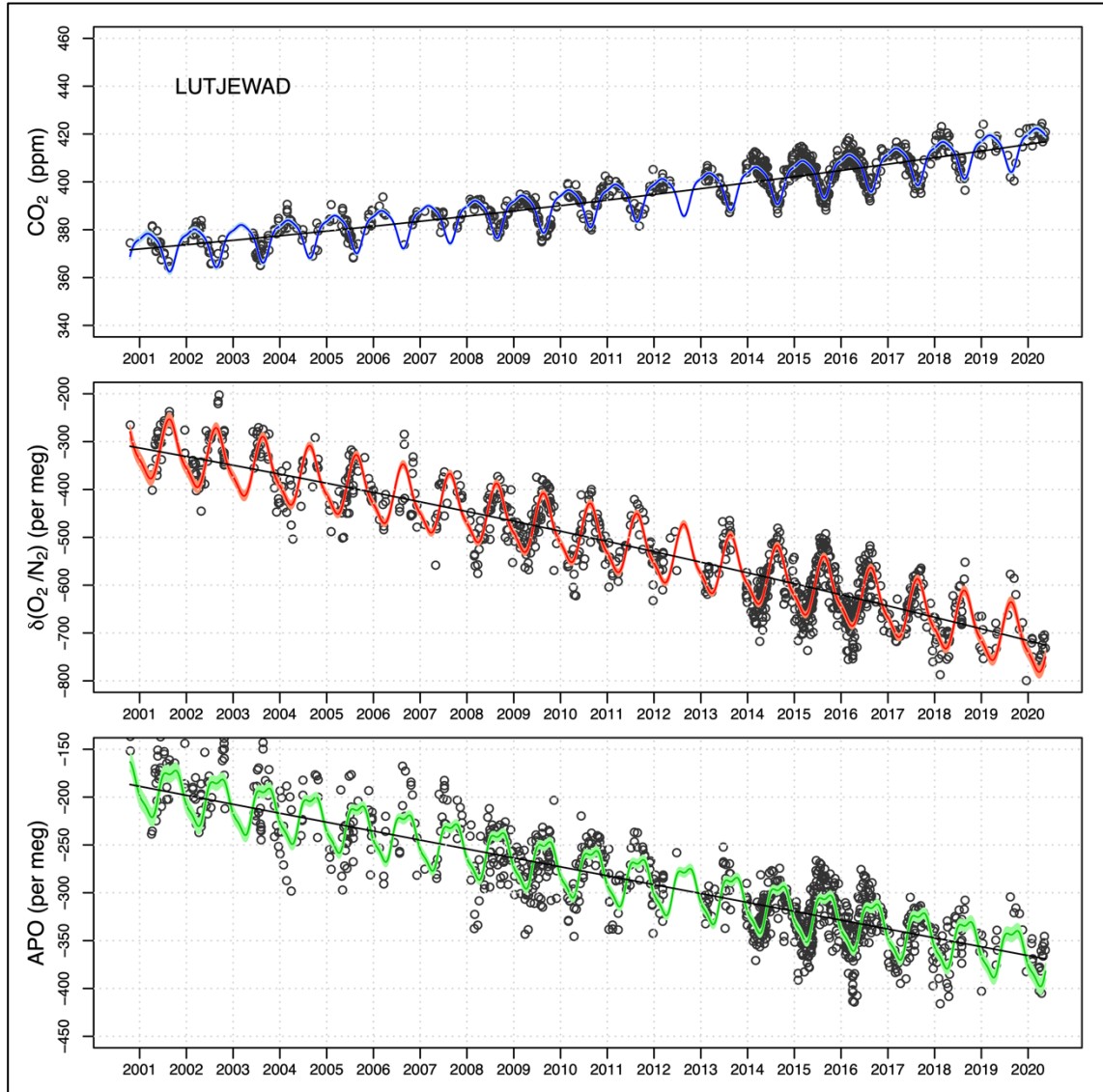

**Figure 7: Flask record from Lutjewad station, showing $CO_2$, $\delta(O_2/N_2)$, and APO measurements from 2000 to 2020. The black points are the individual flask measurements, while the black lines are the long-term trend and the coloured lines indicate the trend with seasonal components derived from the combined quadratic and harmonic regression. The uncertainty ranges (2-sigma) in the fits are indicated by lighter shades of the same colours. For comparability, the y-axes ranges are scaled to represent the 5 per meg : 1 ppm ratio.**

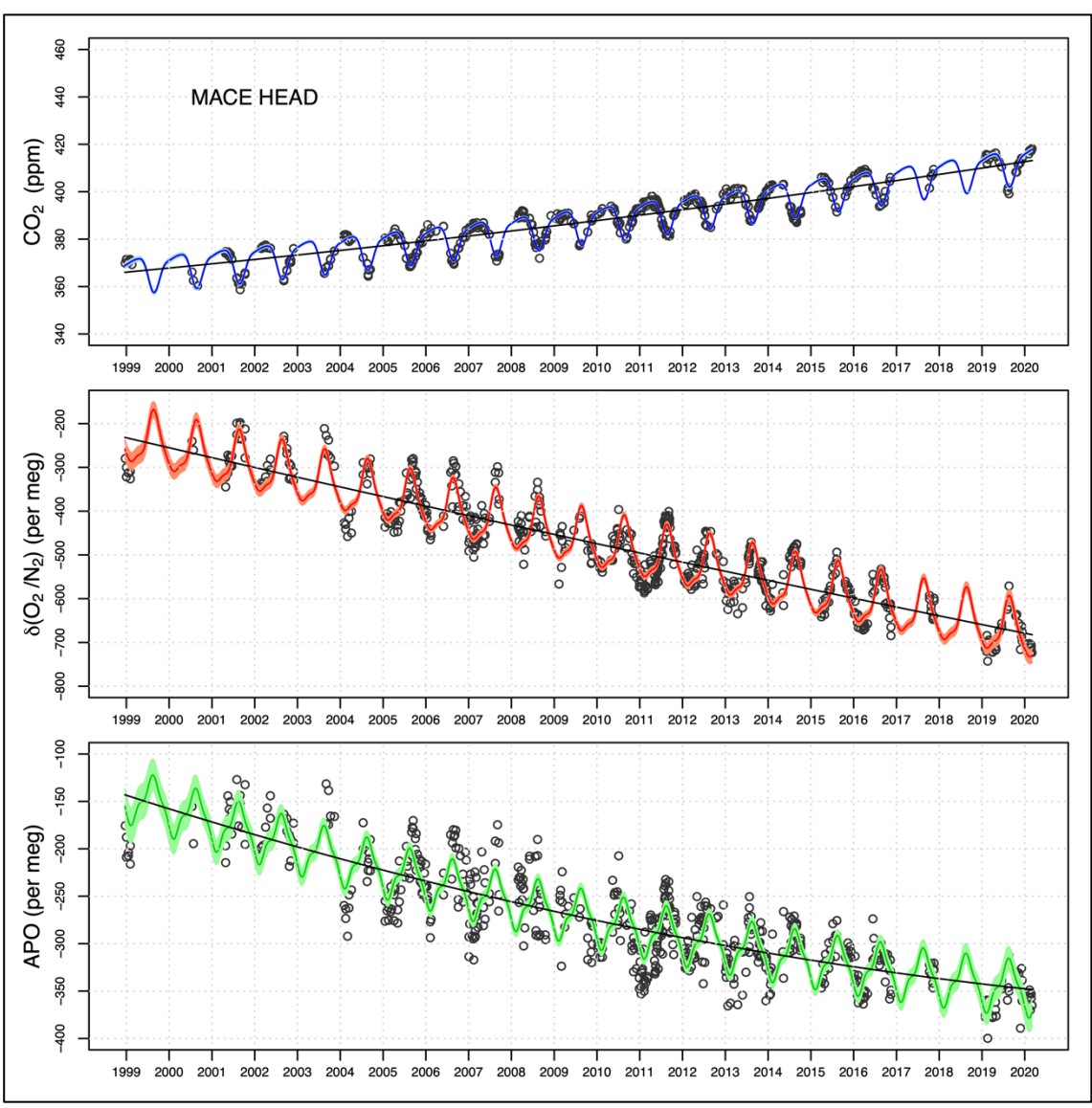

**Figure 8: As for Fig. 7 but for Mace Head station.**

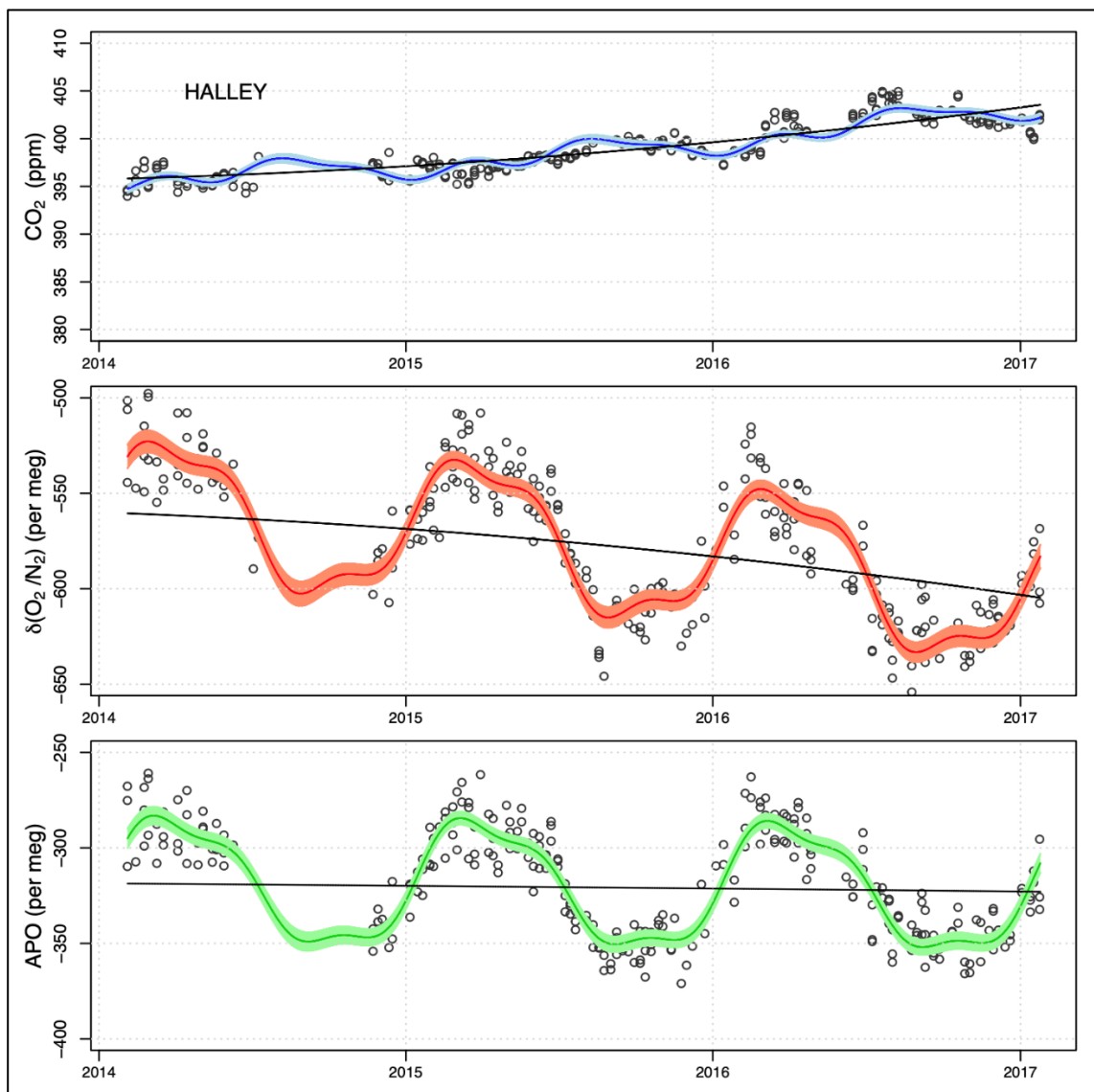

**Figure 9: As for Fig. 7 and 8, but for Halley station and from 2014 to 2017.**

## 4.2 Seasonal cycles

The seasonal cycles of $CO_2$, $\delta(O_2/N_2)$, and APO for all three stations are presented in Fig. 10. The seasonal components
are extracted from the total fits (detrended) and presented as 1-year cycles. In general, the $CO_2$ seasonal cycles at
Lutjewad and Mace Head are similar in size and shape, although the average seasonal amplitude is higher at Lutjewad
($16.8 \pm 0.5$ ppm) than Mace Head ($14.8 \pm 0.3$ ppm). The $CO_2$ seasonal cycle at Halley station, on the other hand, has a
much smaller amplitude of $3.0 \pm 0.3$ ppm, as is generally the case for the ocean-dominated Southern Hemisphere due
to the absence of a terrestrial biosphere influence. Lutjewad and Mace Head show very similar, and significantly higher
$\delta(O_2/N_2)$ seasonal amplitudes ($131 \pm 6$ per meg and $130 \pm 6$ per meg, respectively) than that at Halley ($76 \pm 4$ per meg),
due to the influences of the terrestrial biosphere. In APO this influence is cancelled because APO is invariant to
terrestrial biosphere processes, and the Halley amplitude is even somewhat higher than that of Lutjewad and Mace
Head ($65 \pm 3$ per meg compared to $54 \pm 4$ and $61 \pm 5$ per meg, respectively). All numerical seasonality parameters of
the three stations are given in Table 4 below.

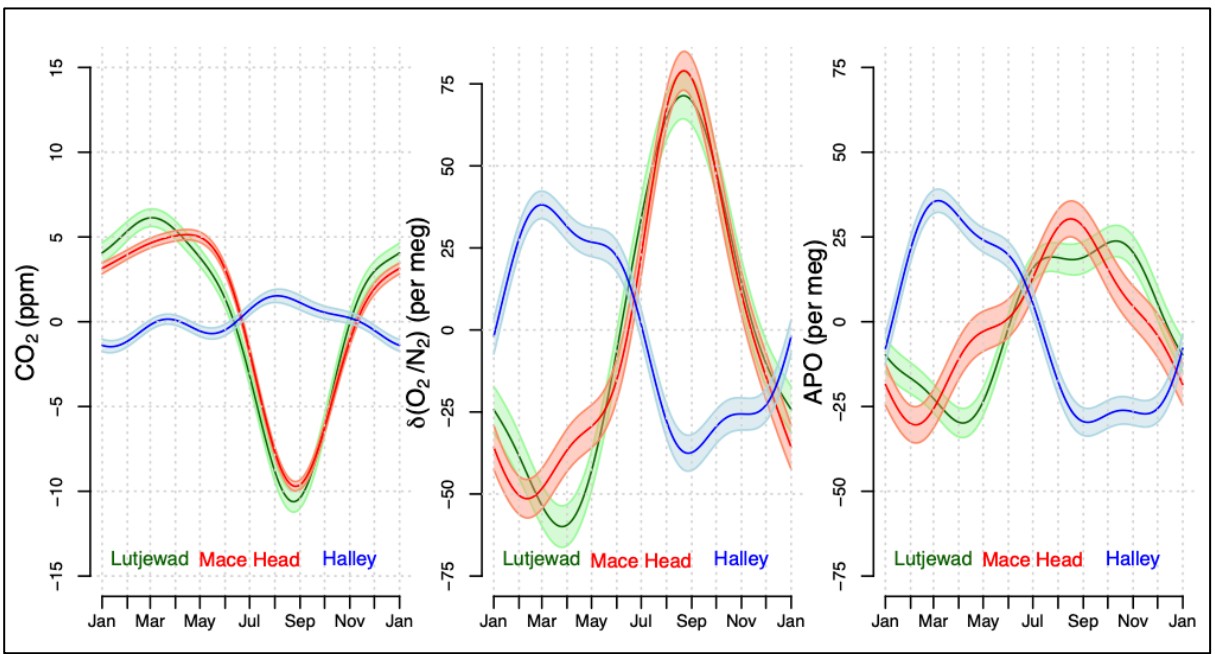

**Figure 10: The detrended average seasonal cycles of CO₂ (left panel), δ(O₂/N₂) (middle panel), and APO (right panel) of stations Lutjewad (plotted in green), Mace Head (plotted in red), and Halley (plotted in blue). The uncertainty margins (2-sigma) in the fits have been indicated by lighter shades of the same colours.**

**Table 4: Trend and seasonality fit parameters of the measurement records from all three stations, as presented in Fig. 7-9**

|  |  | Lutjewad (2002-2018) | Mace Head (2002-2018) | Halley (2014-2017) |
|---|---|---|---|---|
| $CO_2$ | Seasonal amplitude (ppm) | $16.8 \pm 0.5$ | $14.8 \pm 0.3$ | $3.0 \pm 0.3$ |
|  | Average trend (ppm/year) | $2.31 \pm 0.07$ | $2.22 \pm 0.04$ | $2.60 \pm 0.20$ |
|  | Day of min. value | 236 (Aug 24th) $\pm 13$ | 238 (Aug 26th) $\pm 11$ | 11 (Jan 11th) $\pm 12$ |
|  | Day of max. value | 62 (Mar 3rd) $\pm 26$ | 105 (Apr 15th) $\pm 30$ | 216 (Aug 4th) $\pm 14$ |
| $\delta(O_2/N_2)$ | Seasonal amplitude (per meg) | $131 \pm 6$ | $131 \pm 6$ | $76 \pm 4$ |
|  | Average trend (per meg/year) | $-21.2 \pm 0.8$ | $-21.3 \pm 0.9$ | $-15 \pm 3$ |
|  | Day of min. value | 85 (Mar 26th) $\pm 23$ | 42 (Feb 11th) $\pm 33$ | 239 (Aug 27th) $\pm 18$ |
|  | Day of max. value | 234 (Aug 22nd) $\pm 19$ | 234 (Aug 22nd) $\pm 13$ | 59 (Feb 28th) $\pm 21$ |
| APO | Seasonal amplitude (per meg) | $54 \pm 4$ | $61 \pm 5$ | $65 \pm 3$ |
|  | Average trend (per meg/year) | $-9.3 \pm 0.5$ | $-9.7 \pm 0.9$ | $-1.4 \pm 2.4$ |
|  | Day of min. value | 96 (Apr 6th) $\pm 21$ | 38 (Feb 7th) $+ 30$ | 250 (Sep 7th) $\pm 12$ |
|  | Day of max. value | 284 (Oct 11th) $\pm 29$ | 229 (Aug 17th) $\pm 29$ | 66 (Mar 7th) $\pm 17$ |

**5 Discussion**

**5.1 Measurements at Lutjewad, Mace Head, and Halley**

Here, we discuss our measurement records in more detail. At first, the difference in the progression of trends in δ(O₂/N₂) and APO between Lutjewad and Mace Head (Fig. 7 and 8) suggests that there could be an issue with the flask sampling procedure at Mace Head, such as the way the samples are dried. At Lutjewad, the sampling process has been more

closely controlled thanks to the vicinity of our laboratory enabling frequent visits, multiple tests and other measurements taken from the same sample lines. Furthermore, a comparison of the Lutjewad data with data from the nearby Weybourne coastal station in the UK (presented in Sect. 5.2) showed very good agreement. As both Lutjewad and Mace Head samples share the same measurement procedure, measurement and calibration issues cannot explain their differences, so the differences must either be real, or related to the flask sampling procedure. It takes longer to transport the flasks from Mace Head to Groningen than from Lutjewad and thus contaminations of the samples through the valve caps might have occurred. For the samples from the Halley station, the transport time is even longer, but here, additional protective caps (glass or aluminium) with Viton O-rings are used on the valve caps of the flasks to create small buffer volumes that slow down permeation effects. We tested the preservation of the samples using the protective caps by sending flasks to Halley station that were pre-filled with air of known composition, without actually using them. Back in Groningen, we could conclude the integrity of the samples by comparing the measurements before and after shipment, and we found no significant change in $\delta(O_2/N_2)$ after 26 to 51 months. We found a small drift of 0.4 per meg in $\delta(O_2/N_2)$ after 48 months; and a drift of -0.3 ppm in $CO_2$ after 24 months, on a set of 20 flasks. These numbers would only amount to biases of 0.008 per meg /month in $\delta(O_2/N_2)$ and 0.013 ppm/month in $CO_2$. Unfortunately, the protective caps were not applied to Mace Head samples. Still, it is hard to imagine how such permeation effects could cause a deviating long-term trend in the data given that the flasks were filled to ambient pressure. Furthermore, the time between taking the sample and analysing was a few months at most. If anything, one would expect more scatter in the record. The same holds for sampling problems, such as incomplete drying.

To summarise, the trends at Lutjewad are as expected while those at Mace Head are not, so if there are no systematic sampling errors, the differences in $\delta(O_2/N_2)$ and APO in Mace Head compared to Lutjewad might be partially caused by the sparse and irregular sampling frequency at Mace Head or technical issues that remain undiagnosed. However, it is also worthwhile to consider effects that may be caused by real environmental differences between the two stations. Two effects come to mind: the first is a difference in fossil fuel use (both in quantity and type), which would influence $\delta(O_2/N_2)$ and to a lesser extent also APO. The average fossil fuel exchange ratio (ER) for the Netherlands, when accounting for all fossil fuel types, is $1.60 \pm 0.02$ for the 2000-2020 period, much higher than that for Mace Head (1.49, see van Der Laan-Luijkx et al. (2010) and the $CO_2$ release and Oxygen uptake from Fossil Fuel Emission Estimate (COFFEE) database by (Steinbach et al., 2011)), and the global average value for all fossil fuel emissions (of 1.38), as also mentioned by Sirignano et al. (2010) and van Der Laan-Luijkx et al. (2010). However, it is unlikely that this is the main explanation of the difference between the two records. Firstly, because at Lutjewad, sampling was selective so as to avoid continental (and thus local fossil fuel) influences as much as possible, and second, because a difference in trends would need a gradual change in the ER. Data from Statistics Netherlands (CBS, 2021) show that the ER of the Netherlands has changed by no more than 0.02 over the period 2000-2020, too small to be of influence on the observed difference in the trends at Lutjewad and Mace Head. The next potential (though less likely) cause for differences between Mace Head and Lutjewad are changes in North Atlantic oxygen ventilation (Keeling and Manning, 2014) to which the Mace Head observations are more sensitive. Such changes would influence $\delta(O_2/N_2)$ and APO, but not $CO_2$. This is consistent with the fact that the $CO_2$ trends of Mace Head and Lutjewad agree, whereas there are differences in $\delta(O_2/N_2)$ and APO. Changes in the oxygen inventory of the North Atlantic have been reported by Stendardo and Gruber (2012) and Montes et al. (2016) and a relationship with the North Atlantic Oscillation (NAO) has been reported. Data obtained from the NOAA Climate Prediction Center (https://www.cpc.ncep.noaa.gov/data/teledoc/nao.shtml) show that the NAO exhibited gradual changes over the period 2000-2020, from a noisy, more or less balanced positive-negative pattern in the first decade, through to a negative phase in the years 2010-2011 towards gradually mostly positive values for the period 2013-2019. Other potential explanations could include a shift in atmospheric transport or

also data artefact(s). As our operation continues, the coming years might shine light on what are likely or less likely causes.

When comparing the seasonal cycles of the three stations, we can see that while $CO_2$ and $\delta(O_2/N_2)$ seasonal amplitudes at Halley are significantly smaller than those at Lutjewad and Mace Head, the APO seasonal amplitude is slightly higher, agreeing with the model simulation by Tohjima et al. (2012) that the APO seasonal variations in the Southern Hemispheric ocean are larger than those in the Northern Hemisphere due to larger air-sea $O_2$ exchange. As mentioned in Sect. 2, APO values also contain a small influence from fossil fuels, however, by selecting for flasks based on the background conditions, we eliminate as much as possible this influence, especially for the Lutjewad record. As such, our APO values from these three stations represent mostly ocean influences.

As an illustration of the usefulness of the $\delta(O_2/N_2)$ measurement, we calculated the partitioning of $CO_2$ uptake by the terrestrial biosphere and the ocean from the observations at Lutjewad, using the measurements of $CO_2$ and APO concentrations from 2002 to 2018, following the method described by Keeling and Manning (2014), but using the fitted trend lines from Lutjewad instead of global averaged values. This partitioning is illustrated in Fig. 11.

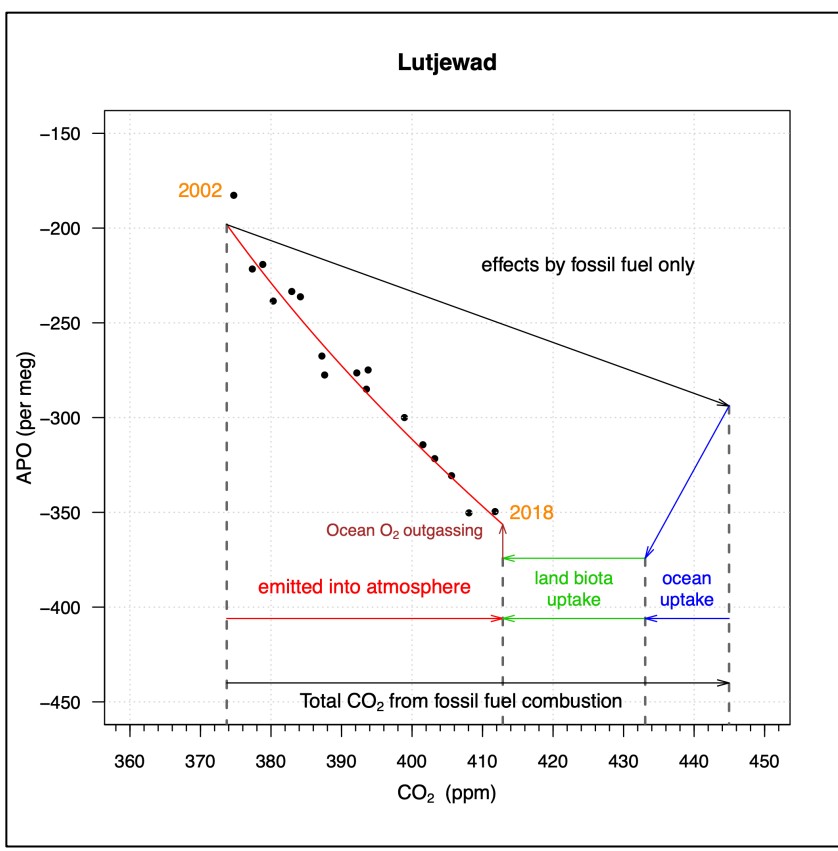

**Figure 11: Vector diagram presenting the calculation of the global land biotic and oceanic carbon sinks for the 2002-2018 period. The black points are the annual averages of the measured APO and $CO_2$ values at Lutjewad, calculated from January to December of each calendar year. The black arrowed line represents the changes in the atmospheric APO and $CO_2$ values that would have occurred if all $CO_2$ emitted from fossil fuel combustion remained in the atmosphere. The ocean uptake is presented by blue arrows and its slope is fixed to the APO/$CO_2$ molar ratio of 1.1 (that represents the removal of the biosphere signal in the definition of APO). The land biota uptake (green) is a horizontal line, as APO does not include a biosphere signal. The ocean $O_2$ outgassing effect is plotted in brown. The red line is a simple trend fitted through the period.**

The black points are the annual averages of the de-seasonalised measurements of APO and $CO_2$ mole fractions at Lutjewad for the period 2002-2018. For calculating the partitioning of fossil fuel $CO_2$, we use equations (2) to (10), and the ocean $O_2$ outgassing component (Z) of $0.44 \pm 0.45 \cdot 10^{14}$ mol yr$^{-1}$ (equivalent to an effect on the carbon sinks of $0.46 \pm 0.48$ PgC yr$^{-1}$) from Keeling and Manning (2014). Furthermore, we use the total fossil fuel emissions for the years 2002-2018 of $8.9 \pm 0.5$ PgC yr$^{-1}$ as derived from the Global Carbon Budget 2021 by Friedlingstein et al. (2021),

and the ER for globally averaged fossil fuel combustion of 1.43 from Jones et al. (2021). To allow comparison of our $\delta(O_2/N_2)$ derived carbon budget to Friedlingstein et al. (2021), we need to adjust our estimate for the river flux of carbon of 0.61 PgC yr$^{-1}$, similar to their $fCO_2$ estimates and inverse results, since all of these methods are based on contemporary observations (see also Hauck et al. (2020)). Using the Lutjewad measurement of $\delta(O_2/N_2)$ and $CO_2$, we then derive for the period 2002-2018 a global land biotic sink (B) of 1.9 ± 1.1 PgC yr$^{-1}$, a global ocean sink (O) of 2.1 ± 0.8 PgC yr$^{-1}$, and the $CO_2$ remaining in the atmosphere amounts to 4.89 ± 0.15 PgC yr$^{-1}$. These values agree well with those reported by Friedlingstein et al. (2021) for the same period: 1.6 ± 0.9 PgC yr$^{-1}$ for B (including emissions from land-use changes) and 2.5 ± 0.4 PgC yr$^{-1}$ for O. The value for atmospheric component A at Lutjewad is slightly higher than the reported average value of 4.66 ± 0.02 PgC yr$^{-1}$ for the 2002-2018 period, so therefore our sum of O and B is lower than that of Friedlingstein et al. (2021) by the same amount. Additionally, we tested the sensitivity of the calculated sinks to different ER values. When applying an ER value of 1.38 (as was used by Keeling and Manning (2014)), the B and O values from Lutjewad record change to 1.5 PgC yr$^{-1}$ and 2.5 PgC yr$^{-1}$, respectively, which is even in better agreement with Friedlingstein et al. (2021). This shows the importance of knowing the ER of the fossil fuel mix and its changes over time in high level of detail in carbon budget calculations using atmospheric measurements of $\delta(O_2/N_2)$ and $CO_2$.

The challenges in making $O_2$ measurements have presented themselves clearly in this work: the sensitivity of the mass spectrometer that require intensive calibration; the quality maintenance of the internal calibration scale to make sure that our measurements can be reported with sufficient quality on the international scale; and the unexpected patterns (especially in APO for Mace Head) that could not be fully explained, partly due to the lack of consistent sampling frequency before 2004 (for both stations), during 2012 (for Lutjewad) and between 2017 and 2019 (for Mace Head). The trend and seasonality fitting procedure are also of great importance, as these are also highly sensitive to irregular sampling frequency and biases in the timing in which the majority of the samples is collected. Nevertheless, our flask measurement records of Lutjewad, Mace Head, and Halley have proven to be informative and valuable in evaluating APO, and with future technical improvement (especially regarding the sampling frequency and the quality maintenance of our internal scale), they will be extended further. In the near future, in addition to more regular sampling frequency at Lutjewad and Mace Head, we aim to improve the frequency at which we perform the measurements on the SIO primary standard cylinders, and also to purchase new primary standard cylinders from them, to produce higher precision conversion to the SIO scale. We also aim to employ more WTs as the current ones are either running out or experiencing considerable noise (see WT 4845 in Fig. 3). We have now added another cylinder to measure along with our last stable WT, to ensure the continuation of our calibration scale quality. More protective measures to the flasks, such as using additional caps or switching to another type of valve, will also be considered, to reduce the risks of potential leakages, permeations, and contaminations during storage and transportation.

**5.2 Comparison with other long-term records**

In Table 5, we compare the seasonal amplitudes of our $CO_2$, $\delta(O_2/N_2)$, and APO measurements with those of some other stations worldwide. As can be seen, the measurements for all three species at Lutjewad and Mace Head agree well with the measurements conducted at other Northern Hemisphere stations Weybourne (UK), Sendai (Japan), and Ny Ålesund (Norway). In the Southern Hemisphere, our $\delta(O_2/N_2)$ and APO measurements for Halley station show an excellent agreement with those at the Syowa station. On the other hand, our $CO_2$ measurements exhibit a much larger and noisier seasonal cycle, which is caused by small leaks during sampling (the details of which are given at the end of this section). Nonetheless, the general concurrence with these stations helps to consolidate the quality of our measurements.

**Table 5: Comparison of the seasonal amplitudes of $CO_2$, $\delta(O_2/N_2)$, and APO at various locations in the world**

| Station | Time period | Latitude | $CO_2$ (ppm) | $\delta(O_2/N_2)$ (per meg) | APO (per meg) | Reference |
|---|---|---|---|---|---|---|
| Ny Ålesund, Spitsbergen | 2001-2010 | 79°N | 15.2 ± 0.4 | 129 ± 4 | 52 ± 3 | Ishidoya et al. (2012b) |
| Weybourne, UK | 2008-2015 | 53°N | 15.2 ± 1.1 | 130 ± 8 | 51 ± 6 | (Barningham, 2018) |
| Lutjewad, the Netherlands | 2002-2018 | 53°N | 16.8 ± 0.5 | 131 ± 6 | 54 ± 4 | This paper |
| Mace Head, Ireland | 2002-2018 | 53°N | 14.8 ± 0.3 | 130 ± 6 | 61 ± 5 | This paper |
| Sendai, Japan | 1999-2012 | 38°N | 13.9 ± 2.5 | 128 ± 22 | 52 ± 10 | Ishidoya et al. (2012a) |
| Syowa station, Antarctica | 2001-2010 | 69°S | 1.1 ± 0.04 | 70 ± 4 | 64 ± 4 | Ishidoya et al. (2012b) |
| Halley station, Antarctica | 2014-2017 | 75°S | 3.0 ± 0.3* | 76 ± 4 | 65 ± 3 | This paper |

*The $CO_2$ seasonal amplitude at Halley is most likely incorrect, details are given at the end of this section.

Additionally, we compare our long-term measurement record with an extended record of Weybourne station (Fig. 12), the first part of which has been published by Pickers (2016) and Barningham (2018). The figure shows the continuous Weybourne record as hourly averages. In general, the two records agree well, except for the period of late 2018 to the end of 2019, when flask measurements (and the fit curves) of $CO_2$ and APO at Lutjewad are slightly higher than those at Weybourne. This difference is due to the fact that the Weybourne hourly measurements make year-to year variability (in trend and seasonal cycle) visible, whereas the Lutjewad record, due to its sparser sampling character, is fitted with a smooth trend and a seasonal cycle that is fixed over the years. Apparently, the 2018-2019 period deviated from the average trend and/or seasonal cycle. However, the overall agreement further consolidates the quality of our measurements.

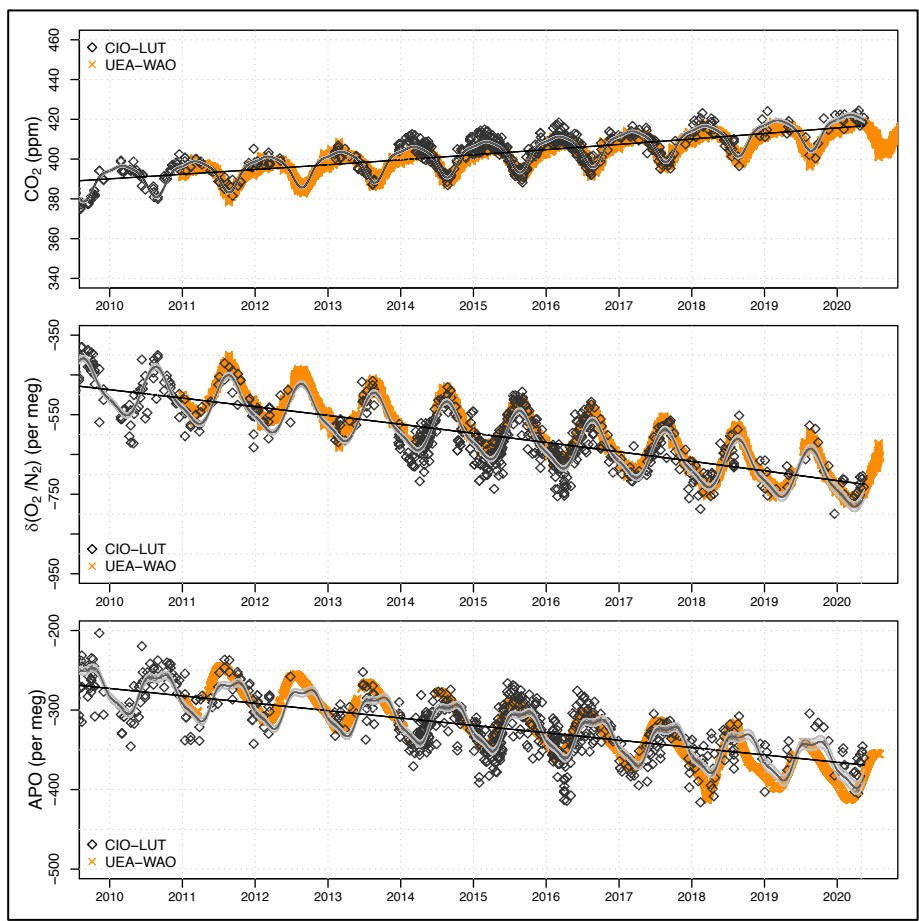

**Figure 12: Measurements of $CO_2$, $\delta(O_2/N_2)$, and APO at Lutjewad (black diamonds) and Weybourne (orange crosses) from 2010 to 2020. The black line and curve are the trend and the combined fit for Lutjewad, respectively. The grey shadings are the 95% CI associated with the total fit.**

For Halley, we compare our $CO_2$, $\delta(O_2/N_2)$, and APO measurements with those conducted by UEA (Fig. 13) (Barningham, 2018). APO measurements between our laboratory and UEA show good agreement, while $CO_2$ measurements show unexpected discrepancies in March, April, and June until August of 2016. $\delta(O_2/N_2)$ measurements also show a slight disagreement, but it is less visible due to a large seasonal cycle and higher scatter. Because APO agrees well, we conclude that the $CO_2$ and $\delta(O_2/N_2)$ anomalies were most likely caused by a small inwards leak when

the flask samples were collected at the station. Laboratory air with higher $CO_2$ mole fractions and lower $\delta(O_2/N_2)$ ratios due to human breathing, probably leaked in. An additional indication pointing to this is that the $CH_4$ and CO mole fractions from the same flasks agree very well with long-term flask measurements made at Halley by NOAA (NOAA, 2021) (not shown here). Such leaks do not influence APO, as the ER from human breathing is close to the value of 1.1 used for the exclusion of the biosphere signal in APO. To better check how much these anomalies would have affected

our measurements, we also use the long-term flask measurements made at Halley from the NOAA website (https://gml.noaa.gov/dv/iadv/graph.php?code=HBA&program=ccgg&type=ts.), since the UEA's measurement period is too short to make a reliable comparison. For $CO_2$, we perform the trend and seasonality fitting procedure, the same as for our own measurements. The measurements between NOAA flasks and UEA agree very well, showing the reliability of UEA's measurements. Thus, the disagreement of $CO_2$ and $\delta(O_2/N_2)$ measurements between our laboratory

and UEA firmly indicate the presence of leakages during March-August 2016, possibly due to human breathing. As aforementioned, APO should be unaffected by these leakages, as can be seen in the agreement between our APO measurements. In the early 2014 period, there are also some anomalies in $CO_2$ measurements as compared to NOAA's, but since there is no available information on $\delta(O_2/N_2)$, we combine NOAA's $CO_2$ measurements with our own $\delta(O_2/N_2)$ measurements to calculate APO. Plotted in blue are the results using the NOAA's $CO_2$ measurements. A clear bias in

APO is visible coinciding with the $CO_2$ anomalies: the $CO_2$ anomalies are around 2 ppm, which would lead to corresponding changes of 10 per meg in APO (since the APO is constructed from "clean" $CO_2$ and "contaminated" $\delta(O_2/N_2)$). The short-term variations in $\delta(O_2/N_2)$ and APO are greater than 10 per meg, masking the suspected leaks. However, the significant difference between the average values for our APO measurements and the ones calculated using the NOAA's $CO_2$ and our $\delta(O_2/N_2)$ (indicated by the black and blue lines in the APO plot, respectively), suggesting that our flasks must have been contaminated with inside air in the early 2014 period.

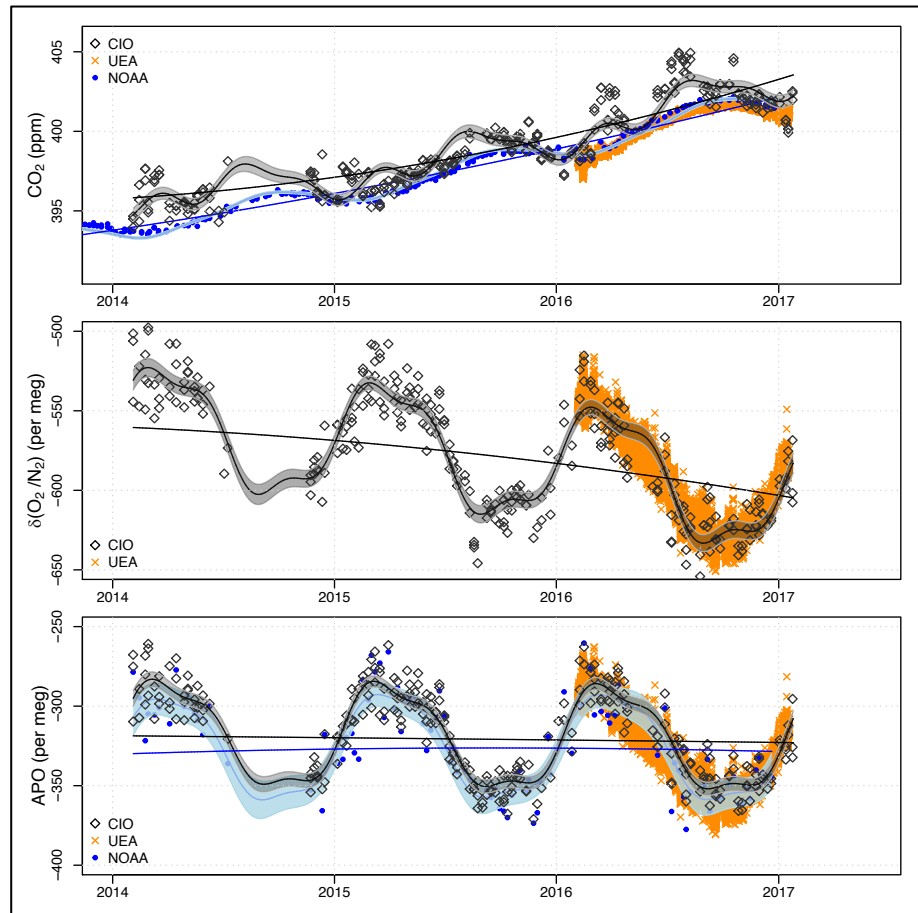

**Figure 13: Measurements of $CO_2$, $\delta(O_2/N_2)$, and APO at Halley conducted by CIO (black diamonds) from 2014 to 2017, and continuous measurements conducted by UEA (orange crosses) in 2016. The black lines and curves are the trends and the combined fit for measurements by CIO, respectively. The grey shaded area is the 95% CI associated with the total fit. The blue points are the in-situ continuous measurements at Halley, taken by NOAA. The blue lines and curves are the trends and the combined fits for the continuous measurements, with the lighter blue shaded area the 95% CI associated with the total fit. The black and blue lines in the APO plot are the average values for our APO measurements, and the ones calculated using the NOAA $CO_2$ and our $\delta(O_2/N_2)$, respectively. The latter is significantly lower, corroborating our conclusion that our $CO_2$ measurements must have been contaminated with inside air (human breathing). The $CO_2$ scale is zoomed in to show the anomalies in 2016 more clearly.**

## 6 Conclusion

We have presented 20-year flask measurement records for $\delta(O_2/N_2)$, $CO_2$ and APO from Lutjewad and Mace Head, along with 3-year records from Halley. We also presented results of the calibration procedures of our instruments. Due to the sensitive nature of oxygen measurements, we conducted an extensive and intensive calibration procedures, which demonstrated a long-term stability for $\delta(O_2/N_2)$ of 3 per meg in 11 years based on our own internal cylinders and 8.6 per meg in 10 years based on our Scripps primary standards. Measurements of the global primary standard cylinders (from SIO) and inter-comparison cylinders (from the Cucumber and GOLLUM programmes) consolidate the stability, quality, and comparability of our calibration procedure, although there are some indications that our calibration scale might not be entirely stable over the past 20 years. However, the results from those various programmes are not

consistent, and therefore inconclusive. The long-term records from Lutjewad and Mace Head provided useful information on the two-decadal trends and seasonality of $CO_2$, $\delta(O_2/N_2)$, and APO, showing good agreements with other stations around the world, especially the Weybourne Atmospheric Observatory in the UK. We found long term trends during the period 2002-2018 of $2.31 \pm 0.07$ ppm yr$^{-1}$ for $CO_2$ and $-21.2 \pm 0.8$ per meg yr$^{-1}$ for $\delta(O_2/N_2)$ at Lutjewad, and $2.22 \pm 0.04$ ppm yr$^{-1}$ for $CO_2$ and $-21.3 \pm 0.9$ per meg yr$^{-1}$ for $\delta(O_2/N_2)$ at Mace Head. The notable differences in the year-to-year progression of $\delta(O_2/N_2)$ and APO trends between Lutjewad and Mace Head might in part be caused by the sparse sampling frequency at Mace Head, but also may potentially be indications of influences from the changes in continental fossil fuel use, different degrees of sensitivity to the North Atlantic $O_2$ ventilation, a shift in atmospheric transport, or an artefact in the data. Using the measurements at Lutjewad for 2002-2018, the partitioning of atmospheric $CO_2$ sinks into the global terrestrial biosphere and the oceans are $1.9 \pm 1.1$ PgC yr$^1$ and $2.1 \pm 0.8$ PgC yr$^{-1}$, respectively. These values agree well with the numbers reported in the most recent Global Carbon Budget. The Halley record shows that the APO seasonal variations in the Southern Ocean are slightly larger than those in the Northern Hemisphere due to larger air-sea $O_2$ exchange there, and illustrates clearly the influences of oceanic processes on the variations in APO and atmospheric $O_2$. With better maintenance of our internal scale, more regular sampling frequency, and better quality-control of the sampling process, the reliability of our future flask measurements will be improved.

**Data availability**

The accompanying database comprises three csv files. The files contain the information on the $CO_2$, $\delta(O_2/N_2)$, and APO measurements (measured values and associated uncertainties) of the three stations, and are named after the corresponding station and the measured parameter (9 files in total).

All files are published by the ICOS Carbon Portal, and are available at https://doi.org/10.18160/qq7d-t060 (Nguyen et al., 2021).

The additional data presented in this paper are available upon request.

**Author contributions**

LNTN, HAJM, and ITL conducted the data analyses, produced all figures and tables, and wrote the manuscript. ITL and HAJM designed the methodology and framework for the calibration procedure of the DI-IRMS. BAMK conducted the technical work and prepared the flask samples at Lutjewad station, and carried out the $CO_2$ and $\delta(O_2/N_2)$ measurements from flasks collected at all three stations. HAS calibrated the $CO_2$ data at CIO. AEJ, NB, and TB performed the measurements at Halley station, prepared the flask samples, produced the data for comparison. PAP and ACM provided the data from Weybourne station. All co-authors contributed to the writing of the manuscript.

**Competing interests**

The authors declare that they have no conflict of interest.

**Acknowledgements**

We would like to thank our colleagues and collaborators at CIO (Janette J. Spriensma for logistics, Henk Jansen for help with the Optima DI-IRMS, Marcel de Vries for the Halley sampler construction, and Ramon R. Richie for help with the measurement database). Furthermore, we thank the collaborators at the stations in collecting and transporting the flask samples, specifically T.G. Spain at Mace Head, and the overwintering teams at Halley station. We thank Eric
755 J. Morgan (SIO) for updating us with new values for the primary standard cylinders. We are also grateful to the UEA staff and students (Michael Pateki, Philip Wilson, Grant Forster, Anh Dieu Tran and Leigh Fleming) and the Atmospheric Measurement and Observation Facility at the UK National Centre for Atmospheric Science (NCAS-AMOF), for kindly providing data from Weybourne. We also thank the three referees for taking their valuable time to provide us with very detailed feedback, upon which we could further complete our work. The work at Lutjewad received
partial financial support over the years by the Dutch Research council (NWO), the European Union Integrated Project Carbo-Ocean (511176) and the Dutch national CATO-2 programme. Atmospheric $O_2$ and $CO_2$ measurements at Weybourne were supported by UK Natural Environment Research Council (NERC) grants NE/F005733/1, NE/I013342/1, QUEST010005 and NE/S004521/1. The Weybourne atmospheric $O_2$ and $CO_2$ measurements have been supported by the National Centre for Atmospheric Science (NCAS), funding agreement R8/H12/83/037 since
December 2013 onwards. I.T.L. received funding from the Dutch Research Council (NWO) (Veni grant 016.Veni.171.095). A.E.J and N.B. were supported by the BAS core programme "Polar Science for Planet Earth". T.B. and P.A.P. were supported by UK NERC studentships, NE/L50158X/1 and NE/K500896/1. P.A.P. and A.C.M. have received support from the NERC-funded DARE-UK (Detection and Attribution of Regional greenhouse gas Emissions in the UK) project, grant agreement no. NE/S004211/1.

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
