# Peer review of "Two decades of flask observations of atmospheric $\delta(O_2/N_2)$ , $CO_2$ , and APO at stations Lutjewad (the Netherlands) and Mace Head (Ireland), and 3 years from Halley station (Antarctica)"

_Earth System Science Data, 2021_

## Referee Comment (RC1)

**Review's comments**

**Manuscript Number:** *ESSD-2021-213*

**Title:** Two decades of flask observations of atmospheric δ(O₂/N₂), CO₂, and APO at stations Lutjewad (the Netherlands) and Mace Head (Ireland), and 3 years from Halley station (Antarctica)

**Authors**: Nguyen, L. N. T., Meijer, H. A. J., van Leeuwen, C., Kers, B. A. M., Scheeren, H. A., Jones, A. E., Brough, N., Barningham, T., Pickers, P. A., Manning, A. C., and Luijkx, I. T.

This paper provides two decadal data of atmospheric $CO_2$ and $O_2$ observed at Lutjewad and Mace Head and 3-year record at Halley. As is the case with the atmospheric $CO_2$, the atmospheric $O_2$ data from a variety of laboratories are also expected to be synthetically analyzed by using atmospheric transport models, biogeochemical models, and so on. However, compared with the atmospheric $CO_2$ measurements, the atmospheric oxygen measurements are still very challenging because we need much more efforts in the process of the air sampling, storing, analysis, and scale maintenance. Especially, it is crucially important to how to keep the $O_2$ scale stability. The authors describe the details of the calibration procedure and several efforts to check the $O_2$ scale stability. However, the authors should make much more effort to clarify the data quality and quantitatively describe the uncertainties associated with the flask measurements in the manuscript. Although I found that the paper contains material that should be published in ESSD, I recommend the manuscript to be published after following minor revisions.

**General comments:**

I understand authors' various effort to keep highly precise measurements of the atmospheric $O_2/N_2$ ratio of the flask samples. However, the δ(O₂/N₂) and APO values at Lutjewad and Mace Head plotted in Figs 6 and 7 show rather scattered plots, which don't seem to be real variations. So, I suspect that the uncertainty of the flask measurement is not so small to adequately detect the atmospheric variation. It is

crucially important to clarify the total uncertainty associated with the $O_2$ data of the flask samples for the synthetic analyses together with the data from other laboratories. Nevertheless, I cannot find any clear description of the analytical precision and the repeatability of the flask measurements in the manuscript. In addition, the authors described the contamination of the flask samples collected at Halley during the storing period. If the same type of the flasks were used for the air sampling at Lutjewad and Mace Head, there is a possibility that the contamination would cause the positive and negative biases of the $CO_2$ and $O_2/N_2$ values for the flask samples, respectively. These potential biases should be also evaluated in the manuscript.

The $O_2$ scale stability is also very important as the authors also recognized. Although the evaluation of the stability is very difficult because there is no absolute scale at present, the authors should quantitatively evaluate the overall stability of the $O_2$ scale in this manuscript. In section 3.1, the authors described that the standard deviations of the repeated measurements of the working tanks were less than 13.5 per meg. Did it mean that the uncertainty of the CIO scale stability was estimated to be about 1 per meg/yr (=13.5 per meg/14 years)? Probably, the results of the COLLUM cylinders would also give a clue of the quantitative evaluation of the $O_2$ scale stability. The evaluation of the uncertainty of the scale stability is directly related to the evaluation of the uncertainty of the carbon budget evaluation described in page 20.

I think that the trend of APO at Mace Head is rather curious because the decreasing rate of APO trend gradually decrease from -15.15 per meg yr$^{-1}$ in 2002 to -5.83 per meg yr$^{-1}$ in 2018. The authors attributed to the $O_2$ emissions from North Atlantic associated with the gradual changes of the NAO. However, I cannot accept the mechanism that the $O_2$ emissions from the North Atlantic only influenced the $O_2/N_2$ and APO at Mace Head. The APO decreasing rates are computed from the fitted quadratic functions. However, taking the data variability for both sites and sparse sampling frequency for Mace Head into account, I suspect that there are no significant differences in the trends between the two sites.

**Specific comments:**
Page 2, line 55: "(Tohjima, 2005)" should be "(Tohjima et al., 2005)". And please add

the coauthors to the reference (Page 29, line 827).

Page 3, line 102-114: At Lutjewad, the air sample was dried by passing it through a Nafion drying tube. How about the Mace Head and Halley stations? The Nafion drying tubes were used at both sites?

Page 4, line 123: It would be better to clarify the temperature of the cryogenic drier.

Page 4, line 132-136: Were the same 2.5-liter glass flasks as Lutjewad used at Mace Head and Halley stations?

Page 8, line 252: Is it possible to describe the linear function to convert the $O_2/N_2$ value based on the CIO scale to that based on SIO scale? Is the conversion function fixed during the observation period of this study? In addition, I think it would be better to describe the uncertainties for the coefficients of the linear function. Such data would be useful to consider the propagation errors for the $O_2/N_2$ values of the flask samples and the standard cylinders.

Page 8, line 267-268: WT4845 show rather unstable $O_2/N_2$ values. It would be informative, if possible, to describe the reason of the instability.

Page 9, line 298-299: I believe that the conversions of the CIO values to the SIO values are based on the fixed conversion function. If so, the discrepancies in the $O_2/N_2$ ratio between the assigned values and the measured values suggest that systematic change in the conversion function, which correspond to the change in the CIO scale, or change in the $O_2/N_2$ values in the Scripps primary cylinders.

Page 9, line 303-305: Do the authors mention that the linear conversion function is often calibrated based on the measurements of the Scripps primary standards as shown in Fig. 4?

Page 10, line 317-318: "Manning et al., 2015" is not in the list of References.

Page 11, line 340: Why do the authors refer to the "WMO extended compatibility goal of 10 per meg"? The extended compatibility goal is set for the studies like urban observations that are strongly influenced by local fluxes. I believe that the authors aim to observe the background air through their three sites observation because they evaluated the global carbon budgets based on their observations in Section 5.1. Therefore, I think the authors should refer the "WMO Network compatibility goal of 2 per meg" here.

Page 11, line 349: All of the GOLLUM cylinders show the increasing drift (Fig. 5). Nevertheless, the authors described that the average overall drift rate significantly small ($4 \pm 6$ per meg yr$^{-1}$). How did the authors calculate the uncertainty of 6 per meg yr$^{-1}$? In addition, I think that the scale drift rate of 4 per meg yr$^{-1}$ is not small because it corresponds to bias of 1.6 PgC yr$^{-1}$ for the carbon budget calculation.

Page 11, line 349-350: The WMO compatibility is defined as "a measure of the persistent bias between measurement records". Thus, it should not be compared with the scale drifting rate.

Page 12, line 370-372: If the authors fit a combination quadratic function and three harmonics to the data by using a least square method and do not use a digital filtering method of Thoning et al. (1989), the authors don't need to refer to Thoning et al. (1989).

Page 13, line 392-394: I don't understand the reason the exclusion of the last 2 years data. For example, the data at Mace Head in 2017 are much sparser than the data in the last year (2019). Additionally, I cannot accept the authors' idea that the sparse data in the last two years introduce biases in the fits. Since there are enough data to determine the average seasonal cycle for both Lutjewad and Mace Head, the larger number of data, even if the sparse, can cause the better fitting results.

Page 13, line 403-405: Please see the comment for Page 13, line 392-394.

Page 13, line 415: I think the longest period for the trend calculation is 17-year (from

2002 to 2018).

Page 18, line 483-486: It should be better to describe the detail of the drying method at Mace Head because there is no description.

Page 19, line 534-537: For the comparison of the observed results between Lutjewad and Mace Head, it is enough to simply compare their trends. I think there is no need for the authors to examine much about carbon budget calculations.

Page 19, line 536: The calculation method adopted in this study is not exactly same as that of Keeling and Manning (2014). In Keeling and Manning (2014), the NOAA's global mean $CO_2$ data was used to evaluate the accumulated $CO_2$ in the atmosphere and globally averaged annual mean APO estimated from the limited background observations was used to evaluate the change in the APO (not fitted trend line).

Page 20, Figure 10: There is no explanation about the red lines in the figures. In addition, the exact period for each annual average shown by the black dot is unclear. Is the period of the annual average for 2002 from January 2002 to December 2002 or from July 2001 to June 2002?

Page 21, line 583: I cannot understand the meaning of "noisier seasonal amplitude".

Page 21, line 596-597: Does the trend for Weybourne show much faster decrease than that for Lutjewad? It would be better to plot the trend for Weybourne in the figures.

Page 22, line 604: The authors described that the $CO_2$ discrepancies are shown in "the first half of 2016". But I think that the discrepancies are shown in July and, probably, August 2016. It should be clarified.

Page 22, line 603-607: It would be better to show the correlation plot of the differences of the flask $CO_2$ and $O_2/N_2$ from the continuous observations. The slope of the scatter plot would give us the information about the origin of the contamination.

Page 22, line 601: I cannot find "NOAA, 2021" in the list of References.

Page 22, line 618-621: The contamination of the flask samples collected at Halley is clearly shown from the comparison of $CO_2$ mole fractions of the flask samples with the in-situ continuous data.

---

## Author Response (AR1)

Ref 1:

This paper provides two decadal data of atmospheric $CO_2$ and $O_2$ observed at Lutjewad and Mace Head and 3-year record at Halley. As is the case with the atmospheric $CO_2$, the atmospheric $O_2$ data from a variety of laboratories are also expected to be synthetically analyzed by using atmospheric transport models, biogeochemical models, and so on. However, compared with the atmospheric $CO_2$ measurements, the atmospheric oxygen measurements are still very challenging because we need much more efforts in the process of the air sampling, storing, analysis, and scale maintenance. Especially, it is crucially important to how to keep the $O_2$ scale stability. The authors describe the details of the calibration procedure and several efforts to check the $O_2$ scale stability. However, the authors should make much more effort to clarify the data quality and quantitatively describe the uncertainties associated with the flask measurements in the manuscript. Although I found that the paper contains material that should be published in ESSD, I recommend the manuscript to be published after following minor revisions.

Thank you for these comments, we have implemented changes to better clarify the uncertainties following the suggestions throughout the manuscript.

**General comments:**

I understand authors' various effort to keep highly precise measurements of the atmospheric $O_2/N_2$ ratio of the flask samples. However, the $\delta(O_2/N_2)$ and APO values at Lutjewad and Mace Head plotted in Figs 6 and 7 show rather scattered plots, which don't seem to be real variations. So, I suspect that the uncertainty of the flask measurement is not so small to adequately detect the atmospheric variation. It is crucially important to clarify the total uncertainty associated with the $O_2$ data of the flask samples for the synthetic analyses together with the data from other laboratories. Nevertheless, I cannot find any clear description of the analytical precision and the repeatability of the flask measurements in the manuscript. In addition, the authors described the contamination of the flask samples collected at Halley during the storing period. If the same type of the flasks were used for the air sampling at Lutjewad and Mace Head, there is a possibility that the contamination would cause the positive and negative biases of the $CO_2$ and $O_2/N_2$ values for the flask samples, respectively. These potential biases should be also evaluated in the manuscript.

We have added in the uncertainty from the flask measurements, and also the total uncertainty associated with the final long-term trends. As for the potential contamination, it is unlikely to cause significant biases on the flask samples from Lutjewad and Mace Head. We did a storability test on flasks going to Antarctica, where we pre-filled a set of flasks that then went to Antarctica, stored there for ~2 years before coming back to our lab for re-measurements. We found a negligible drift of 0.4 per meg in dO2/N2 after 48 months; and a drift of -0.3 ppm in CO2 after 24 months, on a set of 20 flasks. These numbers would only amount to biases of 0.008 per meg /month in dO2/N2 and 0.013 ppm/month in CO2. We collected our flasks from Lutjewad weekly, and Mace Head monthly, therefore the systematic

effects (if any) would only be negligible. Only leakages during each individual sampling session would give rise to outliers, and they should be discarded during our filtering process.

The O$_2$ scale stability is also very important as the authors also recognized. Although the evaluation of the stability is very difficult because there is no absolute scale at present, the authors should quantitatively evaluate the overall stability of the O$_2$ scale in this manuscript. In section 3.1, the authors described that the standard deviations of the repeated measurements of the working tanks were less than 13.5 per meg. Did it mean that the uncertainty of the CIO scale stability was estimated to be about 1 per meg/yr (=13.5 per meg/14 years)? Probably, the results of the COLLUM cylinders would also give a clue of the quantitative evaluation of the O$_2$ scale stability. The evaluation of the uncertainty of the scale stability is directly related to the evaluation of the uncertainty of the carbon budget evaluation described in page 20.

Our O2 scale stability is determined by the stability of our long-term WTs (standard deviation of 13.5 per meg in 14 years) and that of our Scripps primary standard cylinders (8.6 per meg in 10 years). We have now added this information in our main text.

I think that the trend of APO at Mace Head is rather curious because the decreasing rate of APO trend gradually decrease from -15.15 per meg yr$^{-1}$ in 2002 to -5.83 per meg yr$^{-1}$ in 2018. The authors attributed to the O$_2$ emissions from North Atlantic associated with the gradual changes of the NAO. However, I cannot accept the mechanism that the O$_2$ emissions from the North Atlantic only influenced the O$_2$/N$_2$ and APO at Mace Head. The APO decreasing rates are computed from the fitted quadratic functions. However, taking the data variability for both sites and sparse sampling frequency for Mace Head into account, I suspect that there are no significant differences in the trends between the two sites.

It is indeed, from the long-term calculation, that there is minimal differences in trends between Lutjewad and Mace Head. However, due to some unknown reasons – which we can only attribute speculatively to some possibilities – there are significant differences in the annual trends (i.e. the gradient of the fit curve).

**Specific comments:**

Page 2, line 55: "(Tohjima, 2005)" should be "(Tohjima et al., 2005)". And please add the coauthors to the reference (Page 29, line 827).

We have fixed the reference

Page 3, line 102-114: At Lutjewad, the air sample was dried by passing it through a Nafion drying tube. How about the Mace Head and Halley stations? The Nafion drying tubes were used at both sites?

We have added the drying agents at Mace Head ($Mg(ClO_4)_2$, the same as in Halley). Nafion is only used at Lutjewad.

Page 4, line 123: It would be better to clarify the temperature of the cryogenic drier.

We added in the temperature (although it is already described at line 94).

Page 4, line 132-136: Were the same 2.5-liter glass flasks as Lutjewad used at Mace Head and Halley stations?

We added the information (and yes, they are all of the same type of flask).

Page 8, line 252: Is it possible to describe the linear function to convert the $O_2/N_2$ value based on the CIO scale to that based on SIO scale? Is the conversion function fixed during the observation period of this study? In addition, I think it would be better to describe the uncertainties for the coefficients of the linear function. Such data would be useful to consider the propagation errors for the $O_2/N_2$ values of the flask samples and the standard cylinders.

We included the function now. The function is fixed, based on the measurements of the Scripps primary standard cylinder that have been corrected for drifts for the whole period.

Page 8, line 267-268: WT4845 show rather unstable $O_2/N_2$ values. It would be informative, if possible, to describe the reason of the instability.

Unfortunately, we don't know what was wrong with the WT4845, but it might be related to the fact that its value is rather low in comparison to the other tanks – which suggests that the cylinder might contain contaminated air or there could be leaks on the pressure reducer.

Page 9, line 298-299: I believe that the conversions of the CIO values to the SIO values are based on the fixed conversion function. If so, the discrepancies in the $O_2/N_2$ ratio between the assigned values and the measured values suggest that systematic change in the conversion function, which correspond to the change in the CIO scale, or change in the $O_2/N_2$ values in the Scripps primary cylinders.

The conversion of CIO to SIO is indeed a fixed linear function, based on all of the measurements of the SIO cylinders over time. The differences between the assigned and measured values are minimized in this function, and to our opinion there is no unambiguous indication to assume a change in this function over time.

Page 9, line 303-305: Do the authors mention that the linear conversion function is often calibrated based on the measurements of the Scripps primary standards as shown in Fig. 4?

We have added the fixed function that we use, so the function is not calibrated often, but the tank are measured and have been used in the fixed function spanning the whole period presented in the paper.

 "Manning et al., 2015" is not in the list of References.

It is now properly changed to Manning et al 2015, in the reference list.

 Why do the authors refer to the "WMO extended compatibility goal of 10 per meg"? The extended compatibility goal is set for the studies like urban observations that are strongly influenced by local fluxes. I believe that the authors aim to observe the background air through their three sites observation because they evaluated the global carbon budgets based on their observations in Section 5.1. Therefore, I think the authors should refer the "WMO Network compatibility goal of 2 per meg" here.

We have adjusted the comparison to the WMO network compatibility.

 All of the GOLLUM cylinders show the increasing drift (Fig. 5). Nevertheless, the authors described that the average overall drift rate significantly small (4 ± 6 per meg yr$^{-1}$). How did the authors calculate the uncertainty of 6 per meg yr$^{-1}$? In addition, I think that the scale drift rate of 4 per meg yr$^{-1}$ is not small because it corresponds to bias of 1.6 PgC yr$^{-1}$ for the carbon budget calculation.

The uncertainty of 6 per meg/yr is based on individual drifts of each GOLLUM cylinder. And indeed, the drift is not small, but significantly smaller than the uncertainty 11 ± 18 per meg of Cucumbers. The comparison between GOLLUM and Cucumbers is just to show that there is no clear indication of a significant drift in our scale. We have updated the text accordingly.

 The WMO compatibility is defined as "a measure of the persistent bias between measurement records". Thus, it should not be compared with the scale drifting rate.

We have changed it, and removed the comparison.

 If the authors fit a combination quadratic function and three harmonics to the data by using a least square method and do not use a digital filtering method of Thoning et al. (1989), the authors don't need to refer to Thoning et al. (1989).

We have referred to Thoning et al. 1989 as the basis of the use of this function, but we added that we do not use the digital filtering.

 I don't understand the reason the exclusion of the last 2 years data. For example, the data at Mace Head in 2017 are much sparser than the data in the last year (2019). Additionally, I cannot accept the authors' idea that the sparse data in the last two years introduce biases in the fits. Since there are enough data to determine the average seasonal cycle for both Lutjewad and Mace Head, the larger number of data, even if the sparse, can cause the better fitting results.

There was a significant problem with our DI-IRMS for the end of 2019 until all of 2020 that affected the quality of our measurements, so the best we can include is the first ¾ of 2019.

Page 13, line 403-405: Please see the comment for Page 13, line 392-394.

Please see above

Page 13, line 415: I think the longest period for the trend calculation is 17-year (from 2002 to 2018).

We have changed it to 17 years now.

Page 18, line 483-486: It should be better to describe the detail of the drying method at Mace Head because there is no description.

We have added the drying method at Mace Head.

Page 19, line 534-537: For the comparison of the observed results between Lutjewad and Mace Head, it is enough to simply compare their trends. I think there is no need for the authors to examine much about carbon budget calculations.

We think it's still worthwhile to illustrate what information could our data convey, aside from just a long-term trend. Also, reviewer #2 highlights this in the general comments.

Page 19, line 536: The calculation method adopted in this study is not exactly same as that of Keeling and Manning (2014). In Keeling and Manning (2014), the NOAA's global mean $CO_2$ data was used to evaluate the accumulated $CO_2$ in the atmosphere and globally averaged annual mean APO estimated from the limited background observations was used to evaluate the change in the APO (not fitted trend line).

We have now fixed this to reflect the different method.

Page 20, Figure 10: There is no explanation about the red lines in the figures. In addition, the exact period for each annual average shown by the black dot is unclear. Is the period of the annual average for 2002 from January 2002 to December 2002 or from July 2001 to June 2002?

We have added in a description of the red line. The period of annual average is from January to December of each year.

Page 21, line 583: I cannot understand the meaning of "noisier seasonal amplitude".

We have changed this to seasonal cycles.

Page 21, line 596-597: Does the trend for Weybourne show much faster decrease than that for Lutjewad? It would be better to plot the trend for Weybourne in the figures.

WAO data is unfiltered, so there are a lot of non-background data points in this record. We did not manage to update the record to only background conditions as of yet, and are looking for possibilities with the co-author in charge. However, we would not like to postpone the re-submission due to this issue.

Page 22, line 604: The authors described that the $CO_2$ discrepancies are shown in "the first half of 2016". But I think that the discrepancies are shown in July and, probably, August 2016. It should be clarified.

We have fixed this to be more specific.

Page 22, line 603-607: It would be better to show the correlation plot of the differences of the flask $CO_2$ and $O_2/N_2$ from the continuous observations. The slope of the scatter plot would give us the information about the origin of the contamination.

There is no continuous $\delta O_2/N_2$ measurements at Halley so it is not possible to plot the differences between flask and continuous $\delta O_2/N_2$.

Page 22, line 601: I cannot find "NOAA, 2021" in the list of References.

It's there but presented differently since it is a website. It is fixed now.

Page 22, line 618-621: The contamination of the flask samples collected at Halley is clearly shown from the comparison of $CO_2$ mole fractions of the flask samples with the in-situ continuous data.

Yes, we agree, and we changed the text to reflect this.
* * *
Ref 2:
In this paper, the authors present 20 years of observational $\delta(O_2/N_2)$ and $CO_2$ data obtained at three ground-based stations. They also present a detailed description of the calibration procedures of their $\delta(O_2/N_2)$ scale over 15 years. The $\delta(O_2/N_2)$ scale was confirmed to be stable enough to estimate global ocean and land $CO_2$ sinks based on the long-term trends in the observed $\delta(O_2/N_2)$ and $CO_2$. It is important to validate the global $CO_2$ budget, reported by Global Carbon Project, using independent estimations such as those reported in this study. Therefore, the dataset is a valuable contribution to a better understanding of the global carbon cycle. However, I have found some issues that need to be addressed before publication. These are listed below. In particular, some of the interpretations of the observational results are unwarranted. I understand that the ESSD is a data journal, but I think a substantial discussion is also recommended in the paper, particularly considering the high impact of the journal.

Thank you very much for your review, we have addressed the comments below.

1) Line 61: Tohoku University, Japan should be added as a research organization that continues to make long-term systematic observations of $CO_2$ and $O_2$. Goto et al (https://agupubs.onlinelibrary.wiley.com/doi/full/10.1002/2017JG003845) and/or Ishidoya et al. (https://www.tandfonline.com/doi/full/10.3402/tellusb.v64i0.18964) need to be listed as suitable references.

We have added Goto et al. to the list.

2) Line 126: "(Sturm et al., 2004))" should be corrected to read "(Sturm et al., 2004)".

It is now fixed.

3) Lines 116–142: The descriptive detail of the flask sampling procedure at each of the sites need to be the same. Information about the models of the pump used, as well as about the flow rates, inner pressures of the flask, drying agents, and usage of an aspirated inlet need to be described for all the sites. If the size of the flask is different at each site, for example, then the size information needs to be given.

We have added additional information and made the description more uniform.

4 ) Lines 164-180: The measurement precision of $\delta(O_2/N_2)$ for flask measurements is not shown. Is it the same as the long-term standard deviation of 10.2 to 13.5 per meg for cylinder measurements? Please clarify.

Based on the flask data of LUT and MHD, the measurement precision is 7 to 13 per meg, and is now added to the main text.

5) Line 188 and references: "(Tohjima, 2005)" should be corrected to "(Tohjima et al., 2005)".

This is fixed.

6) Line 197: "(van der Laan-Luijkx et al., 2013)" should be corrected to "van der Laan-Luijkx et al. (2013)".

This is fixed.

7) Chapter 3: Examination of the long-term stability of the $\delta(O_2/N_2)$ scale presented in this chapter is highly detailed. It ensures reliability of the long-term trends in the observed $\delta(O_2/N_2)$. However, I was not able to follow how the authors evaluated the uncertainty in the observed long-term trends caused by the uncertainty of the $\delta(O_2/N_2)$ scale. The authors described "Bilbo and Frodo present a minor drift similarly to that observed by our SIO cylinder 7008 (while the other 2 SIO cylinders did not exhibit this behaviour as shown in Sect. 3.2); and our internal WTs all show no overall drifts, we consider our calibration procedure as sufficient" (lines 355–358). Does this mean that the observed $\delta(O_2/N_2)$ values are determined against "the

other 2 SIO cylinders" and no uncertainty is considered for the long-term trends in the observed $\delta(O_2/N_2)$ associated with the scale's uncertainty? In addition, quantitative information about the uncertainty in the $\delta(O_2/N_2)$ scale during the period prior to 2006 is not provided (line 387–394). Did the author consider the scale's uncertainty before 2006 to determine the long-term trends of the observed $\delta(O_2/N_2)$?

The observations are determined against all 3 SIO cylinders. The conversion of the scales between each different period considers the uncertainties of the measurements in each, so therefore it's reflected in the final uncertainty of the flask measurements. The flask measurements before 2006 therefore had larger uncertainties due to the scale conversion and also affected the long-term trend's uncertainties. The scale uncertainties are included in the calculation of the final uncertainty.

8) Lines 379–381: If the larger fraction of discarded measurements at Lutjewad, compared to those at Mace Head, is related to the effects of local sources/sinks as the authors suggest, then not only $\delta(O_2/N_2)$ but also $CO_2$ would be observed to be more scattered at Lutjewad than Mace Head. Would the authors agree with this? If the scatter is seen only in $\delta(O_2/N_2)$, then it is highly likely that the scatter is due to an artificial fractionation of $O_2$ and $N_2$ rather than due to any of the local effects.

We think the referee has misunderstood the text, as the % discarded are for both $\delta O_2/N_2$ and $CO_2$, not one of them.

9) Lines 495–499: What is the protective cap made of? If the authors confirmed that a permeation effect was reduced significantly by using the cap, then it is valuable to provide a fuller description. Anyway, I agree with the authors that the permeation effect and incomplete drying are not the causes of the significant difference in the long-term trends between Lutjewad and Mace Head.

They are made of glass, we added in this information.

10) Lines 516–524: I think the discussion surrounding the interpretation of the difference in the long-term trends between Lutjewad and Mace Head from the viewpoint of changes in the North Atlantic oxygen ventilation is too speculative. Hamme and Keeling (2008) discussed differences in the interannual variations between the northern and southern hemispheres in relation to the North Atlantic oxygen ventilation (the authors referred to Keeling & Manning (2014), but the original paper on this topic was published by Hamme and Keeling (2008)). However, since both Lutjewad and Mace Head are located on the European continent, the horizontal atmospheric transport is much faster than the meridional transport. Therefore, I expect the contribution of the North Atlantic oxygen ventilation to the interannual variations observed at the two sites would be similar. Do the authors have any supporting information to clarify this issue, such as the simulated results using an atmospheric transport model?

It is indeed speculative since we lack the supporting data to confirm this. We did not check this with a model, so we added in the text that it is a potential cause for differences.

11) Line 552: The ER for globally averaged fossil fuel combustion should be calculated using the latest Global Carbon Budget data (Friedlingstein et al., 2020)

From the Global Carbon Budget data, the averaged ER of 1.434 is now used.

12) Lines 603–621: I think the argument to conclude that the $CO_2$ and $\delta(O_2/N_2)$ anomalies were most likely caused by a small inwards leak is weak. The $CO_2$ values at Halley observed by CIO appear to be higher by about 2 ppm than those obtained by UEA and NOAA, so that the corresponding APO decrease is about 10 per meg by assuming biospheric signal. On the other hand, short-term variabilities of $\delta(O_2/N_2)$ at Halley appear to be larger than 10 per meg, compared to the data from CIO and UEA, and I cannot distinguish systematic difference between them. Therefore, consistency between the APO from CIO and UEA does not provide enough evidence of the small inwards leak. I suspect the increase in $CO_2$ measured by CIO may be due to deterioration of $CO_2$ during the storing period, such as desorption of $CO_2$ from inner wall of the flask or some other effects. I would like to hear the authors' thoughts on this.

If we had he continuous $\delta O_2/N_2$ data at Halley, then this would be immediately clear. However, since we do not have that, we depend on the known $CO_2$ of both records (continuous and flasks) and the known $\delta O_2/N_2$ of the flasks to see the effects on APO. We do agree that the short-term variations may have masked the signals of the suspected leaks, however due to the lack of information, this conclusion appears to be the most probable cause. What we do know is, the effects of storing are not the cause, because we performed storage tests and they show remarkable quality over long period of time: we found a negligible drift of 0.4 per meg in $\delta O_2/N_2$ after 48 months; and a drift of -0.3 ppm in $CO_2$ after 24 months, on a set of 20 flasks. These numbers would only amount to biases of 0.008 per meg /month in dO2/N2 and 0.013 ppm/month in $CO_2$ so that is why leaks are the most probably cause.

Ref 3:

In this manuscript Nguyen et al. present ~20 years of observations of atmospheric oxygen and carbon dioxide from two Northern Hemisphere flask sampling stations and 3 years of observations of the same from an Antarctic station. The authors are presenting a substantial dataset of value to the greater O2 and CO2 communities. The data are openly accessible from the ICOS Carbon Portal, and the files are self-explanatory (with one minor exception). The manuscript is of good quality and I recommend publication after in ESSD after addressing some of the minor points I raise below.

Thank you for your review, we address the comments below.

MAJOR COMMENTS

1. Missing details: Some of the important details pertaining to the measurements are not presented. The reader is instead pointed to some relevant citations. This forces the data user to read this paper, plus 2-3 more, including a PhD thesis. I think reproducing some of the key details here would be a service to the reader, particularly details on the flask design/shape, and more information about the mass spec analysis. I am also surprised there are no relevant changes to mention during 20 years of sampling in either sample collection, analytical approach, gas handling, storage, etc, or changes to the automatic flask sampler. Also useful would be details on the different tanks used. Maybe I missed this, but I did not see anything about the valve type and seal, volume, interior, etc. Could the authors include some kind of change log, or table of notable events? If there is truly nothing to mention, I applaud the authors' consistency over 20 years of sampling!

We have added more information about the sampling procedure at the stations, and the measurements with the DI-IRMS. As for the plumbing diagram and design of the flasks, we think it is better to refer to the cited papers, since they provide detailed information about them.

2. Uncertainty and data quality: There is not much in the way of uncertainty analysis or constraint. I would say this is the biggest shortcoming of this paper. The reader is left with no real guidance as to how to assess the uncertainty in the individual measurements, or, perhaps more importantly, the trend. It does not seem like they have a good handle on the uncertainty due to primary tank drift (e.g. Keeling et al 2006 Tellus 59). A full uncertainty analysis may be out of scope, but they could at least put some constraints with the data at hand on the long-term trend and the reproducibility of a given flask measurement. As I see it, if someone wanted to use these flask records in some kind of analysis, this paper would be the main source of guidance. As such, I hope the authors can provide a bit more help in how an interested user could constrain the uncertainty of the measurements.

We have now added the uncertainties of the flask measurements and the effects on trends.

3. Supporting data: The authors are publishing sample time, sample height, and analyte concentration, but further data on the flasks is not included: analysis date, fill pressure, average flow rate, temperature data for sampling and analysis if it exists, etc. This supporting data would be helpful for anyone interested in further QA/QCing or using the data. The authors also do not include the other species used to filter/select the samples for background conditions. I suggest the authors should also seriously consider supplying the non-background data with flags, instead of only the background samples, and the CO/Radon data used to filter them. Or, at least provide DOIs as to where one could find it.

It is indeed a good idea to include the full raw data, we will do so in the coming time.

MINOR COMMENTS

Data files: It is not clear to me from the paper or the header what exactly the standard deviation column represents in the data files. I suggest the authors add this to the header, or put it in a subsection in the paper describing the files.

We have added the extra information in the updated data files.

L17-18: better to provide a metric here than to use the subjective "high-quality". Also, I am not sure if inter-comparisons tell us anything about the quality of the calibration--all of the labs could be making the same mistake.

That is true, but if at least we can show some consistent results, then it's better than all labs showing very different values.

L19: suggest striking the "internationally-recognised" for the sake of brevity.

It is changed to "international"

L25: Compatible can only be assessed if two measurements are made on the same air (tank or background), so I don't think it is correct to say that seasonal cycles are compatible if measured at different locations. Better to say they are in good agreement.

Indeed, it is changed to "in good agreement" now.

L40 - "a strong aide" -- Curious wording

Valuable is now used instead.

L50-63: suggest cutting this entire paragraph up to "Our Laboratory...", and combining with the next paragraph.

We will leave it as it is now, since it highlights the importance of $O_2$ measurements, and shows previous work on this topic for reference.

L81: What does the "(formerly)" mean? It used to be called this but the name has changed, or it's not operational now? Please clarify.

Indeed it had to cease operation in 2016/7 and moved to a new location due to a crack on the ice shelf.

L94: Nafion driers are not very common in O2/N2 measurements. Does Nafion fractionate O2/N2? If the authors have tested this, I would encourage them to include such results here (or provide a citation).

Except for water, all other species should have negligible gradient over the Nafion membrane, since we supply the outer side of the nafion drier with the exhaust of the system. It is therefore unlikely that anything can cause fractionation of the $\delta O_2/N_2$ values.

L115: Please provide a plumbing diagram(s) of the flask samplers.

Information is added, but as for the plumbing diagrams, we would like to refer readers to Neubert et al 2004 (cited) for much more detailed information.

L116: Could you include a drawing or picture of one of the flasks? Do they have dip tubes?

We have added the information – the flasks have dip tubes.

L123: Dried to what dewpoint? Please include specifications on the cryotraps.

This is already described in line 97, but I also added it in line 123.

L124: What is the flow rate during flask sampling?

This information has been added now.

L124: Atmospheric pressure varies, please give exact fill pressure with observed range.

We have now stated that flasks are filled to "current atmospheric pressure". The exact values vary slightly, but the flasks are always kept at the same pressure as the atmospheric pressure in the laboratory.

L131: Please give full details on sampling protocols for Mace Head.

This information has been added now.

L170: "Relatively very stable" -- ambiguously worded

Relative to most other gases (except noble gases), $N_2$ is very stable. We have removed relatively.

L191: The influence of fossil fuel burning on APO is not small--that is why there is a large trend in APO.

"Small" is now omitted.

L210: Is there a systematic difference between first, second, and third analyses? Why sometimes 2 and sometimes 3?

There are no systematic differences between the duplicates. Usually there are 2, but sometimes when there is an obvious problem with one of the duplicates, we perform a third analysis. We added this to the text.

L215: I don't fully follow -- you are assigning the WT a value and then assigning flasks a value based on comparison with the flask? Or flasks are assigned values from the MREF and then corrected for long-term drift through the WT?

Yes, we measured flasks first as a difference against the MREF, then the MREF are used in combination with the WT to correct for the long-term drifts + changes in scales of the MREF (as can be seen in Fig.3 panel 1, showing the WTs through different MREFs).

L249: From Figure 3 it looks like some of the drift is not well-described by an average drift rate. Can the authors comment on this?

Yes, indeed there are still some small drifts not entirely corrected. We tried our best to eliminate as much drifts as possible but unfortunately some small periods are not as well-defined, which is not fully satisfactory.

L264: I am still a little confused about how values are assigned. The WTs are given a value based on the MREF, and then Equation 3 is applied to the flask samples? If so, wouldn't the WTs by definition have to be stable? Or do you mean that they are stable relative to one another? Do the authors have a comment as to why 4845 is so variable?

Yes, both the WTs and the MREFs are stable, however due to many potentials that could cause drifts (analyser drifts mostly), the "measured" values are not, hence we first corrected for all these drifts presented in a sample by relating all of these changes against a baseline that we chose as our internal baseline scale (i.e. the CIO scale), then from that we convert the measurements of the samples into the SIO by a direct connection that we established between CIO and SIO scale by calibrating with the Scripps cylinders. As for why 4845 was so variable, we think it might be related to the very low value of the cylinder, which suggests potentially contaminated air inside the cylinder or small leakages in the pressure reducer during measurements.

L275: I think this is a little misleading, since changing MREF cylinders leads to large offsets in the record. I agree that generally based on Figure 3 the scale looks stable after the correction, but as I understand what the authors describe they are blind to WT drift. Or?

The WT drifts are already shown in the raw data of the WTs vs MREFs, and we corrected for the drifts by individually separating the record into each individual MREF period, and dividing even smaller within those, to correct for the WT drifts.

L290: But the primaries look systematically low (7002 and 7003), and 7008 shows clear drift. I would strike this sentence ("The ensemble thus suggests...")

If there was a systematic error in our calibration, it would show in all cylinders, and we therefore decided to keep this sentence.

L305: One primary is clearly drifting relative to the other two, does it really make sense to include this tank in the ensemble? Also, one would expect cylinders to drift over time. If possible these effects should be accounted for in an uncertainty analysis.

The drift, while noticeable when comparing to the other 2, only amounts to 1.4 per meg per year. For now, we decide to include it, but in the future, we may purchase new primary cylinders.

L350: It shows drift in your scale only if the GOLLUM cylinders are not drifting. It could be that 7008 is stable and the other two are not, or that all the cylinders are drifting together and 7008 is drifting slightly less or more than them. Without absolute constraints, it is unclear.

That is indeed true that we cannot be 100% sure, but they are the only standards that we have.

L404: "exact multitude of years" -- what does this mean?

It means complete calendar years, i.e. from Jan to Dec.

L465: I think this seasonal cycle section (and section 4.1) is perhaps out of scope for the journal. Suggest to cut.

We politely disagree with this, because the presentation of the data in more details including the seasonal cycles is worthwhile for this journal. We are not sure if the reviewer really means section 4.1, since that section with the main presentation of the data.

L486: "has been under much closer controlled thanks" -- should read e.g. "has been more closely controlled"

It is now fixed.

L492: I do not know what a valve cap is. Surely it is the o-ring which causes the seal? Why would permeation through the o-ring be impacted by an external cap?

The valve cap is just an additional cap (with O-ring) to lessen the potential permeation through the o-rings, as it forms a small buffer volume between flask and outside.

L496: This is great to see, could you include some actual figures or numbers here?

Answer here + The information will be added

L500: I find it hard to believe the trend could be impacted by sampling bias, particularly since Mace Head is sampled/filtered for background conditions.

Not biases in sampling condition per se, but the actual sampling procedure.

L505-510: The decadal trend in APO should be virtually identical between two European background sites, and can't be explained by country-level differences in fossil fuel use. The authors acknowledge this on L509-510. Suggest this be cut.

We agreed, and moved this point into the potential list as suggested below.

L516: I agree that North Atlantic ventilation signals are likely to be present, but there are numerous other possible causes contributing to the different trends visible in APO between the two sites. I think it would be better to simply state in a sentence or two that the authors do not know the cause of the difference between the two stations, and that the list of possible explanations include: changing continental fossil fuel signals, shifts in atmospheric transport, different degrees of sensitivity to North Atlantic ventilation, other possible causes, or an artifact(s) in the data. I think it's important to acknowledge the last one here. I also wonder how much of the curvature at MHD is due to the fit itself--maybe the annual averages aren't actually that different? On this last point, I think calculating the terrestrial and oceanic sinks is out of scope for this paper. I suggest cutting this section and the figures.

Yes, we agree that it is better to suggest a list of causes since there is no conclusive answer for the discrepancies. As for the terrestrial and oceanic sinks, we still want to give them, but then for Lutjewad only, again to illustrate the possible use of the data.

Section 5.2: I also think this is out of scope. The comparison of seasonal amplitudes between sites tells us really nothing about the quality of the data, since we expect there to be station to station differences. It also seems odd to me for an ESSD paper to briefly present data not pertaining to the dataset being published, such as the Weybourne or Halley measurements by other groups. Suggest cutting the entire section.

We would like to politely disagree, since the additional illustrations are still interesting to see. They show what potential information the data carry within them, and how inter-laboratory comparisons are for HAL.

Figure 1: Three more panels showing the local site (e.g. satellite, street view, or topography) would be helpful here.

The most important information about the stations are already included, so we would like to omit these from the figure.

Figure 6 and 7: It is difficult to see the actual data because the fitted curves are on top of the points. I would suggest removing the curves completely and let the data speak for itself. I would also suggest zooming in on the CO2 data a bit more.

We feel the fits really add value to the plots, so we would like to keep them as they are. We chose the scale of $CO_2$ is to match the scale of $\delta O_2/N_2$, to a relative ratio of 1 ppm per 4.8 per meg, so that changes are comparable. We prefer to include the fits.

---

## Referee Report (RR1)

**Review's comments**

**Manuscript Number:** *ESSD-2021-213*

**Title:** Two decades of flask observations of atmospheric $\delta(O_2/N_2)$, $CO_2$, and APO at stations Lutjewad (the Netherlands) and Mace Head (Ireland), and 3 years from Halley station (Antarctica)

**Authors**: Nguyen, L. N. T., Meijer, H. A. J., van Leeuwen, C., Kers, B. A. M., Scheeren, H. A., Jones, A. E., Brough, N., Barningham, T., Pickers, P. A., Manning, A. C., and Luijkx, I. T.

I found the authors revised the manuscript properly in accordance with the most of the reviewers' suggestions and comments. However, I think that there are some ambiguous and/or erroneous descriptions in the revised manuscript. Therefore, I think that several points addressed below should be clarified before acceptance for publication in Earth System Science Data.

**Specific comments:**

Page 9, line 283-286: This paragraph is very important for this study because the stability of the CIO scale is discussed here. However, I think that some additional figure or table should be required to conclude the scale stability of "less than 3 per meg over the 14 years". This is because the differences in the $\delta(O_2/N_2)$ value between WT5279 and WT6168 increased to 7.3 per meg from MREF6170 period to MREF6123 period as listed in Table 2.

Page 15, line 457: Is "20-year period" right? Or "17-year period"?

Page 15, line 474: Is "the COI scale stability (13.5 per meg in 14 years)" right?

Page 21, line 594 (Figure caption): "diagram"

Page 22, line 607-615: If my understanding is correct, the land and ocean sinks reported by Friedlingstein et al. (2021) do not include the riverine flux. The correction of the riverine flux is applied only to the ocean sink estimate based on the ocean $pCO_2$ observations in Friedlingstein et al. (2021) (see section 2.4 Ocean $CO_2$ sink). Additionally, in their study, global ocean biochemistry models (GOBMs) are used to evaluate the anthropogenic ocean sinks, which are the additionally acquired ocean sinks from the natural ocean condition, in which the ocean is considered the $CO_2$ source due to the riverine flux. As the authors discussed in the manuscript, the land and ocean sinks based on the observations of the atmospheric $O_2$ and $CO_2$ do not take into account the riverine $CO_2$ flux. (It is considered that the land biomass is the source of the riverine carbon, which is accompanied by $O_2$ consumption.) Therefore, those fluxes should be directly compared to those reported by Friedlingsteine et al. (2021) without the correction of the riverine flux (0.6 Pg yr$^{-1}$).

Page 22, line 614: "higher" should be "lower".

---

## Author Response (AR2)

**Review's comments**

**Manuscript Number:** ESSD-2021-213

**Title:** Two decades of flask observations of atmospheric $\delta(O_2/N_2)$, $CO_2$, and APO at stations Lutjewad (the Netherlands) and Mace Head (Ireland), and 3 years from Halley station (Antarctica)

**Authors**: Nguyen, L. N. T., Meijer, H. A. J., van Leeuwen, C., Kers, B. A. M., Scheeren, H. A., Jones, A. E., Brough, N., Barningham, T., Pickers, P. A., Manning, A. C., and Luijkx, I. T.

I found the authors revised the manuscript properly in accordance with the most of the reviewers' suggestions and comments. However, I think that there are some ambiguous and/or erroneous descriptions in the revised manuscript. Therefore, I think that several points addressed below should be clarified before acceptance for publication in Earth System Science Data.

Thank you very much for your comments. We hereby address them in details below.

**Specific comments:**

Page 9, line 283-286: This paragraph is very important for this study because the stability of the CIO scale is discussed here. However, I think that some additional figure or table should be required to conclude the scale stability of "less than 3 per meg over the 14 years". This is because the differences in the $\delta(O_2/N_2)$ value between WT5279 and WT6168 increased to 7.3 per meg from MREF6170 period to MREF6123 period as listed in Table 2.

We have now included a graph showing the annual averages of the 3 WTs over the years, along with the fitted trends and the slopes of the trends. From the graph, the year-to-year variability of the cylinders shows the stability of our internal scale better than just the averaged values during different MREF periods.

Page 15, line 457: Is "20-year period" right? Or "17-year period"?

This is now fixed to 17-year period

Page 15, line 474: Is "the COI scale stability (13.5 per meg in 14 years)" right?

This is now fixed to less than 3 per meg in 11 years

Page 21, line 594 (Figure caption): "diagram"

This is now fixed.

Page 22, line 607-615: If my understanding is correct, the land and ocean sinks reported by Friedlingstein et al. (2021) do not include the riverine flux. The correction of the riverine flux is applied only to the ocean sink estimate based on the ocean $pCO_2$ observations in Friedlingstein et al. (2021) (see section 2.4 Ocean $CO_2$ sink). Additionally, in their study,

global ocean biochemistry models (GOBMs) are used to evaluate the anthropogenic ocean sinks, which are the additionally acquired ocean sinks from the natural ocean condition, in which the ocean is considered the $CO_2$ source due to the riverine flux. As the authors discussed in the manuscript, the land and ocean sinks based on the observations of the atmospheric $O_2$ and $CO_2$ do not take into account the riverine $CO_2$ flux. (It is considered that the land biomass is the source of the riverine carbon, which is accompanied by $O_2$ consumption.) Therefore, those fluxes should be directly compared to those reported by Friedlingsteine et al. (2021) without the correction of the riverine flux (0.6 Pg $yr^{-1}$).

We would like to politely disagree with this comment. First of all, the land and ocean sinks as reported by Friendlingstein et al. 2021 are taking into account the adjustment of the riverine flux to derive the $fCO_2$ (previously $pCO_2$) based estimate which feeds into the reported ocean sink, as this is the mean of the GOBMs estimate and the data based $fCO_2$ estimate. Also, the atmospheric inverse results in Friedlingstein et al. 2021 are adjusted in the similar way as the $fCO_2$ estimate. This is because these methods are based on contemporary observations, are therefore it is necessary to remove the pre-industrial ocean source of $CO_2$ to the atmosphere of 0.61 PgC/yr (see also Hauck et al. 2020). The same logic holds for the $\delta(O_2/N_2)$ estimate.

Page 22, line 614: "higher" should be "lower".

This is now fixed.